# The Fermentation Mechanism of Pea Protein Yogurt and Its Bean Odour Removal Method

**DOI:** 10.3390/foods14193363

**Published:** 2025-09-29

**Authors:** Xiaoyue Zhang, Guozhi Ji, Yan Zhao, Bingyu Chen, Wenhui Li, Zimeng Guo, Shan He, András Koris, Xuchun Zhu, Zhishen Mu, Hongzhi Liu

**Affiliations:** 1Key Laboratory of Geriatric Nutrition and Health, Beijing Technology and Business University, Ministry of Education, Beijing 100080, China; 15210132410@163.com (X.Z.);; 2Inner Mongolia Enterprise Key Laboratory of Dairy Nutrition, Health & Safety, Hohhot 011500, China; jiguozhi@mengniu.cn (G.J.);; 3Global R&D Innovation Center, Inner Mongolia Mengniu Dairy (Group) Co., Ltd., Hohhot 011500, China; 4Department of Food Process Engineering, Institute of Food Science and Technology, Hungarian University of Agriculture and Life Sciences, Ménesi út 44, HU-1118 Budapest, Hungary

**Keywords:** pea protein yoghurt, fermentation mechanisms, flavour analysis, bean odour removal method

## Abstract

Pea protein yogurt (PPY), as an alternative to traditional dairy yoghurt, has the advantages of being a green raw material, lactose cholesterol-free, and adaptable to the needs of lactose-intolerant people. PPY was prepared by fermenting a mixture of pea protein and water (1:10, *w*/*v*) supplemented with 5% fructose for 10 h after heat sterilisation. During fermentation, lactic acid bacteria metabolise pea protein to produce aldehydes and other aromatic compounds, imparting a unique sweet–sour balance and mellow flavour. However, issues such as weak gel formation and prominent soybean-like off-flavours severely restrict the development and consumer acceptance of PPY. In this study, five fermentation systems were systematically investigated to elucidate the fermentation mechanisms of pea yoghurt and explore effective methods for eliminating undesirable soy flavours. The results indicated that hydrophobic interactions and disulfide bonds are the predominant forces driving gel formation in PPY. Additionally, the protein content increased by 0.81 g/100 g following fermentation. A total of 43 volatile flavour compounds—including aldehydes, alcohols, acids, ketones, and furans—were identified, among which the concentrations of hexanal and 2-pentylfuran, known markers for soybean off-flavour, significantly decreased. Furthermore, high-temperature and high-pressure treatments (121 °C, 3 min) demonstrated superior effectiveness in reducing soybean-like flavours. Although the high-temperature and high-pressure treatment, double-enzyme hydrolysis, and flavour-masking methods operate through distinct mechanisms, their flavour profiles converged, displaying substantial deodorisation effects and synergistic interactions. These findings provide a theoretical basis and processing parameters for flavour modulation in PPY; however, further formulation optimisation is required to enhance its nutritional and textural properties. PPY shows promise as a potential alternative to conventional dairy products in the future.

## 1. Introduction

Yoghurt, a traditional fermented dairy product highly favoured by consumers for its unique flavour and nutritional health benefits [1], is currently facing significant challenges due to increasing global protein demands. On one hand, carbon emissions have risen with the rapid expansion of the livestock sector, which has now jumped to the top of all sectors; on the other hand, approximately two-thirds of the global population experiences lactose intolerance. Thus, there is an urgent need to identify flavourful, nutritious alternatives to traditional fermented dairy yoghurt. Plant-based proteins, produced globally in large quantities (approximately 4.2 million tonnes per year), are lactose-free, and their products offer a new option for lactose-intolerant people [2,3]. Additionally, they contain zero cholesterol, have higher concentrations of bioactive plant compounds [4], and possess superior environmental sustainability, making them ideal candidates for replacing traditional dairy products [5]. Notably, it is noteworthy that peas are cultivated on a large scale globally, with Russia alone producing approximately 34 million tons annually—accounting for one-third of the world’s total pea production. The Asia-Pacific region has emerged as the fastest-growing market for pea protein consumption, a trend propelled by its large population base and rising overall demand for protein. These significant outputs have garnered considerable attention from both industry stakeholders and researchers. However, peas have defects such as pronounced pea odour, poor solubility, and gelation in food applications, which together limit the development of peas in the food industry.

During traditional yoghurt fermentation, lactic acid bacteria convert glucose and galactose from lactose into lactic acid through the enzymatic action of lactate dehydrogenase. The accumulation of lactic acid reduces the pH of milk, resulting in a sour flavour and triggering the formation of characteristic volatile flavour compounds. Additionally, the reduced pH induces the denaturation of casein proteins, unfolding their peptide chains and neutralising their surface charges, ultimately leading to the formation of a stable gel structure [6]. In contrast, PPY lack lactose and casein proteins, resulting in weaker gelation, lower hardness, and compromised textural properties. Moreover, various volatile compounds generated during the cultivation and processing of legumes [7] impart an undesirable “bean flavour” to soy or pea yoghurts [8,9]. It has been demonstrated that numerous volatile substances produced during milling, protein extraction, and fermentation contribute collectively to the distinct bean flavour of PPY. Among over twenty identified bean-related volatile compounds, hexanal is considered the most critical due to its extremely low odour threshold [10]. These sensory and textural issues significantly affect the production quality and consumer acceptance of PPY, thereby greatly limiting their commercial viability. Consequently, identifying effective interventions to enhance gel formation and mitigate off-flavours in plant-protein yoghurts has become an urgent research priority for the food industry.

Several studies have addressed methods to reduce undesirable bean flavours in PPY. For example, electromagnetic heating has been reported to produce baking aromas capable of masking bean flavours [11]. Low-energy microwave treatment (100 W/g) can mitigate the raw/beany and green/grassy aromas associated with pea protein [12]. Additionally, dual-frequency ultrasound (20 and 40 kHz) treatments have been utilised to disrupt interactions between soy proteins and bean-flavour compounds, thereby facilitating the removal of undesirable aromas [13]. Despite these findings, there remains a research gap concerning the comparative effectiveness of different treatments in reducing bean-flavour compounds specifically in PPY [14,15,16,17]. In this study, we explored the fermentation mechanisms underlying PPY, elucidated the gel network formation processes, and analysed the composition of its volatile flavour substances. Furthermore, we systematically compared the efficacy of various treatment methods in removing bean-flavour compounds from pea yoghurt. The outcomes of this research provide valuable insights and practical references for improving the gel texture and flavour quality of PPY, establishing a theoretical foundation for its further development and commercialisation.

## 2. Materials and Methods

### 2.1. Reagents and Materials

Pea protein powder was obtained from Yantai Shuangta Foodstuffs Co., Ltd., Yantai, China. A mixed leavening agent consisting of Lactobacillus delbrueckii subsp. bulgaricus and Streptococcus thermophilus was supplied by Yiran Biological Co., Ltd. (Shijiazhuang, China). Sugar was purchased from a local supermarket in Beijing, China. The following ortho-ketones (all analytical grade) were sourced from Aladdin: 2-butanone, 2-pentanone, 2-hexanone, 2-heptanone, 2-octanone, and 2-nonanone. All other chemicals used were of analytical grade and obtained from Beijing Chemical Reagent Co., Ltd. (Beijing, China). High-purity nitrogen (99.999%) and 20 mL headspace vials were provided by Shandong Haineng Scientific Instrument Co. Separation was performed using an MXT-5 capillary column (15 m × 0.53 mm, 1.0 μm; Restek, Bellefonte, PA, USA).

### 2.2. Instruments and Equipment

The following instruments were used: a high-pressure homogeniser (JXNANO-15, Shanghai, China); a wall-breaker (BM9, Shenzhen, China); a vertical autoclave steriliser (LDZF-50L-I, Shanghai, China); a thermostatic incubation oscillator (Z-intelligent HNY-301, Tianjin, China); a scanning electron microscope (EmCrafts, Beijing, China); a microwave oven (G80F25MSLVII-ZN(MO), Shenzhen, China); a wall-breaker (BM9, Shenzhen, China); a FlavourSpec^®^ gas-phase ion mobility spectrometer (G.A.S., Dortmund, Germany); a CTC-PAL 3 static headspace autosampler (CTC Analytics AG, Zwingen, Switzerland); and VOCal data processing software (version 0.4.10, G.A.S., Dortmund, Germany).

### 2.3. Preparation of PPY

Pea protein powder was mixed with sterile water at a material–liquid ratio of 1:10; 5% fructose was added; high-pressure homogenisation (60 °C, 30~40 MPa, single instantaneous processing) was carried out; and pea protein solution (PPS) was made using a wall-breaker, a process that included a heating step. Then, the PPS was separated from other solid residues using an 80-mesh strainer to obtain the PPY pre-fermentation solution A1; 0.006% of the fermenter bacterial powder was added for fermentation after being cooled down; fermentation was carried out for 10 h at a temperature of 37 °C; and then, post-fermentation ripening was carried out for 24 h at a temperature of 4 °C to obtain the PPY A2. After the end of storage in 4 °C refrigerator, a low-temperature live-bacteria type of solid yogurt was obtained.

### 2.4. pH, Acidity, and Water-Holding Capacity

The pH of PPY during fermentation was monitored using a pH meter (SANXIN, Shanghai, China). The samples were titrated with 0.1 mol/L NaOH solution, the amount of NaOH solution was recorded, the titratable acidity was calculated, and its value was expressed as °T.

Evaluation of water-holding capacity (WHC) of PPY: the water-holding capacity of PPY was measured according to the method of Delikanli et al. [18]. The mass of an empty centrifuge tube (50 mL) was recorded as m. A sample of PPY (10 mL) was placed in the above centrifuge tube, and the mass was recorded as m1; then, the tube was centrifuged for 10 min at 4 °C, 8000× *g*. The supernatant was removed, and the mass was recorded as m2. The WHC was calculated using the following formula:(1)$$\mathrm{WHC}=\frac{\left(m_{2}-m\right)}{\left(m_{1}-m\right)}\times 100\%$$

### 2.5. Textural Characterisation

Individual texture indices of PPY were determined using a texture meter (Brookfield, Middleboro, MA, USA). The PPY was fermented uniformly using disposable petri dishes so that the thickness was also consistent. Texture conditions: initial speed 2 mm/s, entry speed 1 mm/s, ejection speed 1 mm/s, entry depth 30%, 2 cycles. Measurement indexes: hardness (g), adhesiveness (mJ), cohesiveness, springiness (%), and chewiness (mJ).

### 2.6. Protein Content

Protein content was quantified using the Kjeldahl method as described by Wang et al. [19], with total nitrogen multiplied by a conversion factor of 6.25.

### 2.7. Determination of Anti-Nutritional Factors in PPY

Trypsin inhibitor determination was performed according to GB5009.224-2016, modified.

CaCl_2_-HCl solution: A total of 0.735 g of CaCl_2_ was weighed and dissolved in 1 L of 0.001 mol/L HCl solution, and the pH was adjusted to 3.0 ± 0.1 with 1 mol/L HCl solution and 0.1 mol/L HCl solution.

Trypsin solution (0.0135 mg/mL): Trypsin was placed at room temperature; 13.5 mg of trypsin was weighed into a small beaker, dissolved with calcium chloride hydrochloric acid solution, and transferred to a 1000 mL volumetric flask; and the volume was fixed to the scale with CaCl_2_-HCl solution.

Tris-CaCl_2_ solution: A total of 6.05 g of Tris and 0.735 g of CaCl_2_ dissolved in a 1000 mL beaker were weighed with a scale, with 900 mL of water added beforehand; the pH value was adjusted to 8.2 ± 0.1 with 1 mol/L HCl solution and 0.1 mol/L HCl solution; and water was added until 1000 mL was reached.

L-BAPA solution: A total of 60 mg of L-BAPA dissolved in 1 mL of dimethyl sulfoxide (DMOS) was weighed, transferred to a 100 mL volumetric flask with Tris-CaCl_2_ solution, and diluted to scale. The sample was prepared as it was used.

Determination of trypsin inhibitor activity of samples: The pH of the PPY samples was adjusted to 9.0 with NaOH solution (0.01 mol/L), after which the above samples were centrifuged at 4000 r and 4 °C for 10 min, and after centrifugation, the supernatant was collected and appropriately diluted. As shown in Table 1, certain amounts of L-BAPA solution, Tris-CaCl_2_ solution, trypsin solution, and sample dilution were added into two 10 mL centrifuge tubes and mixed well, and 1 mL of acetic acid solution (5.3 mol/L) was added immediately after holding at 37 °C for 10 min. Then, the tubes were put into centrifuges and centrifuged at 4000 r/min for 10 min. The absorbance of the sample was measured at 410 nm.

### 2.8. Scanning Electron Microscopy (SEM)

The ultrastructure of PPY was assessed by SEM [20]. The PPY samples were freeze-dried in a freeze-dryer, and then, the freeze-dried PPY blocks were fixed on aluminium stakes (10 mm × 10 mm) for SEM using double-sided carbon tape and coated with gold sputtering coatings for 45 s. The samples were imaged at 10.0 kV, and micrographs with a magnification of 50× were recorded for analysis and comparative studies.

### 2.9. Rheological Measurements of PPY

Rheology was measured according to Li et al. [21]. Rheological measurements were performed at 25 °C using a rotational rheometer (Anton Paar Physica MCR 302, Graz, Austria) and a 50 mm diameter plate (PP50). The PPY samples were placed in the centre of the stationary plate, and a thin layer of silicone oil was applied to the periphery to minimise evaporation. The gap between the plates was 1 mm. Steady-state viscosity measurements were made at 1.0% strain and a frequency range of 0.1–100 rad/s. The energy storage modulus (G′), loss modulus (G″), and loss factor (tan δ) calculated from G″/G′ were recorded.

### 2.10. Free Sulfhydryl Groups

Free sulfhydryl groups were determined according to the method of XIONG et al. [22] with slight modifications. The samples were diluted to 2 mg/mL with Tris-Gly buffer (containing 0.086 mol/L Tris, 0.09 mol/L Gly, and 4 mmol/L ethylenediaminetetraacetic acid) at pH = 8. Subsequently, 4.5 mL of diluted sample was reacted with 50 μL of Ellman’s reagent (2-nitrobenzoic acid (DTNB) dissolved in Tris-Gly buffer, mass concentration 4 mg/mL) for 15 min at room temperature in the dark. The solution without DTNB was used as a blank. The absorbance was measured by a spectrophotometer at 412 nm. The free sulfhydryl content was calculated according to Equation (2):Free sulfhydryl content (µmol/g) = 73.53 × A/C(2)
where 73.53 is the molar extinction coefficient (L mol^−1^ cm^−1^), 73.53 = 106/1.36 × 10^4^ (1.36 × 10^4^ is the molar absorptivity, M^−1^ cm^−1^); A is the absorbance at 412 nm; C is the concentration of protein/(2 mg/mL).

### 2.11. Intermolecular Forces in PPY

Intermolecular forces were measured according to the method of Wang et al. [23] with slight modifications. This measurement was carried out using selective buffers (prepared with 0.05 M PBS at pH 7.0). These buffers were able to break certain bonds in PPY, and the different buffer solutions were as follows:

(S1) Ionic bonding: 0.6 mol/L NaCl solution;

(S2) Hydrogen bonding: 1.5 mol/L urea and 0.6 mol/L NaCl solution;

(S3) Hydrophobic forces: 8 mol/L urea and 0.6 mol/L NaCl solution;

(S4) Disulfide bond: 0.5 mol/L β-mercaptoethanol, 8 mol/L urea, and 0.6 mol/L NaCl solution.

Staining solution: mix Kaumas Brilliant Blue solution (1 mL) with PBS (5 mL).

The PPY sample (1 g) was mixed with the above buffer solution (10 mL) and left at 4 °C for 1 h. The mixture was centrifuged at 10,000× *g* and 4 °C for 20 min, and the supernatant was collected. The supernatant (20 μL) was mixed with the staining solution (200 μL) and incubated for 3–5 min. The absorbance was measured at 595 nm. The protein concentration in different buffer solutions was calculated from a standard curve using bovine serum albumin (BSA) as a standard protein, which indirectly gives the intermolecular forces. The solubilities of S1, (S2–S1), (S3–S2), and (S4–S3) represent the contributions of ionic bonding, hydrogen bonding, hydrophobic interactions, and disulphide bonding, respectively.

### 2.12. Different Treatments Before Fermentation and GC-IMS Processing Conditions

#### 2.12.1. Different Treatments Before Fermentation

As shown in Table 2, the preparation process of A2 was consistent with that of Section 2.3; A3 changed the heating of the wall-breaker in Section 2.3 to microwave heating (500 w, 5 min); A4 changed the heating of the wall-breaker in Section 2.3 to autoclaving (121 °C, 3 min); A5 was based on the enzyme digestion of A2 prior to heating in the wall-breaker (alcohol dehydrogenase: 0.5%, aldehyde dehydrogenase: 0.5%, 2 h); A6 was to replace the ingredients with a 1:1 blend of pea protein powder and coconut protein powder; and the other treatments were the same as A2.

#### 2.12.2. Sample Processing Method

A total of 2 g of PPY was weighed in a 20 mL headspace bottle and incubated at 60 °C for 20 min; then, the sample was injected; and 3 sets of each sample were determined in parallel.

#### 2.12.3. Headspace Injection Conditions

Incubation temperature: 60 °C; incubation time: 20 min; injection volume: 500 µL; non-shunt injection; incubation speed: 500 r/min; injection needle temperature: 85 °C.

#### 2.12.4. Gas Chromatographic (GC) Conditions

Table 3 shows the program settings for the PPY GC injection as follows: column temperature: 60 °C; carrier gas: high-purity nitrogen (purity ≥ 99.999%); programmed pressure: initial flow rate 2.0 mL/min for 2 min, linearly increased to 10.0 mL/min within 8 min, linearly increased to 100.0 mL/min within 10 min, and linearly increased to 150.0 mL/min within 10 min; chromatographic runtime: 30 min; injection port temperature: 100.0 mL/min within 10 min. Chromatographic run time: 30 min; inlet temperature: 80 °C.

#### 2.12.5. Ion Mobility Spectrometry (IMS) Conditions

Ionisation source: tritium source (3H); migration tube length: 53 mm; electric field strength: 500 V/cm; migration tube temperature: 45 °C; drift gas: high-purity nitrogen (purity ≥ 99.999%); flow rate: 75.0 mL/min; positive ion mode.

#### 2.12.6. Data Processing

The mixed standards of the six ketones were detected; calibration curves for retention time and retention index were established; and subsequently, the retention index of the substance was calculated from the retention time of the target, which was retrieved and compared with the GC retention index database and the IMS migration time database built into the VOCal software for qualitative analysis of the target.

Reporter, Gallery Plot, and Dynamic PCA plug-ins in VOCal data processing software were used to generate 3D spectra, 2D spectra, difference spectra, fingerprints, and PCA plots of volatile components for the comparison of volatile organic compounds among samples.

### 2.13. Statistical Analysis

All samples were subjected to three replicate measurements, and the results are expressed as mean ± standard deviation. Duncan Multiple Range Test (DMT) was performed using SPSS statistical software (IBM SPSS Statistics, windows software version 20.0) to analyse the differences between the results at *p* < 0.05.

## 3. Results and Discussion

### 3.1. Fermentation Mechanism of PPY

The combination of Streptococcus thermophilus with Lactobacillus bulgaricus subsp. bulgaricus has been shown to produce yogurt with lower synergistic effects, better texture, and superior organoleptic qualities compared to other fermenters in yogurt [24]. After inoculation of Lactobacillus bulgaricus with Streptococcus thermophilus in pea protein solution, they started to utilise the available nutrients for growth and proliferation, producing new proteases [25]. During metabolism, organic acids are rapidly produced, which gradually lowers the pH of the solution, and the proteins begin to pre-aggregate, promoting the formation of gel structures [26]. Once the pH decreases below 6, pea proteins interact and aggregate within the acidic environment, forming a three-dimensional gel network characteristic of PPY. Concurrently, as shown in Table 4, lactic acid bacteria generate numerous flavour-active compounds during fermentation, including acetaldehyde, acetone, acetyl, diacetyl, and acetic acid, imparting the distinctive flavour profile of PPY.

#### 3.1.1. Changes in Protein Content During Fermentation

Protein content is a critical indicator of PPY’s nutritional value, significantly influencing both its quality and health benefits. As shown in Figure 1, the protein content increased with the increase in fermentation time. This is consistent with the results obtained from fermented soybeans by Chaudhary et al. [27]. This may be due to the precipitation of whey at the end of fermentation, which leads to a decrease in water while the total amount of proteins remains the same, resulting in an increase in protein content. This suggests that fermentation promotes higher protein levels, thereby enhancing the nutritional profile of PPY.

#### 3.1.2. Antinutritional Factors in PPY Before and After Fermentation

Trypsin inhibitors are antinutritional factors widely found in plants, particularly in legume seeds. These inhibitors stimulate the pancreas to produce excessive amounts of digestive enzymes, causing an unnecessary loss of endogenous amino acids, potentially leading to pancreatic hyperfunction or, in severe cases, pancreatic hypertrophy [28]. Experimental results demonstrated that the trypsin inhibitor activity in pea protein was initially 0.95 U/g, whereas after fermentation into PPY, it decreased significantly to 0.31 U/g, representing more than a threefold reduction. There are two reasons for this: One is that the acidic environment created by a drop in pH can denature and inactivate trypsin inhibitors [29]. The second reason is that some of the proteases produced by lactic acid bacteria during growth and proliferation can hydrolyse the peptide bonds of trypsin inhibitors, thus destroying their spatial structure and inactivating them [30]. This indicates that fermentation effectively reduces trypsin inhibitor activity, enhancing the nutritional safety of PPY.

#### 3.1.3. Comparative Analysis of the Fingerprints of Volatile Components in PPY Before and After Fermentation

In order to express in more detail the change rule and relative content of volatile substances in different samples under different treatment conditions, we used Gallery Plot plug-in (Tableau Software, Seattle, WA, USA) to draw the fingerprints of volatile substances.

Figure 2 provides a visual comparison of the differences in volatile organic compounds (VOCs) between PPY samples before and after fermentation. Compounds highlighted within the red box in Figure 2 were present at higher concentrations in the pre-fermentation sample (A1), including glutaraldehyde, nonanal, heptanal, cis-4-heptenal, 3-methylbutanal, octanal, 2-pentylfuran, 3-octanol, phenylglyoxal, furfural, hexanal, benzaldehyde, and cis-2-pentenol. Among these, hexanal, 2-pentylfuran, and nonanal are recognised as the primary contributors to the characteristic beany or fishy off-flavour in legume-based products [31]. Notably, flavour compounds responsible for bean odour typically have low sensory thresholds, meaning that even trace amounts can result in perceptible off-flavours [10].

In contrast, compounds within the yellow box in Figure 2 were detected at higher concentrations in the post-fermentation sample (A2). These included 2,3-pentanedione, 3-methylbutyric acid, ethyl azoic acid, 3-methylbutanol, isobutyric acid, 3-methyl-2-butenal, 1-heptanol, 1-hexanol, trans-2-heptenal, 1-butanol, 2-pentanone, (E, E)-2,4-heptadienal, trans-2-octenal, trans-2-pentenal, 1-pentanol, butyraldehyde, ethyl acetate, and 2-butanone. Among these, 2-pentanone is considered a key contributor to the characteristic flavour of traditional PPY. The fermented PPY (A2) exhibited a significant reduction in hexanal, 2-pentylfuran, and nonanal, which are responsible for the pea-like odour. This reduction is likely due to the inactivation of lipoxygenase during heating, which prevents the oxidation of fatty acids that generate these volatile compounds and due to the active enzyme system of lactic acid bacteria during fermentation, e.g., aldehyde dehydrogenase converts the undesirable flavour hexanal to hexanoic acid. Meanwhile, fermentation led to an increase in compounds characteristic of fermented foods, such as 3-methylbutyric acid and isobutyric acid, as well as pleasant aroma components including 1-hexanol, 1-heptanol, and 2-pentanone.

Overall, these findings suggest that fermentation significantly enhances the flavour profile of PPY by reducing undesirable beany off-flavours while promoting the formation of desirable volatile compounds. This is consistent with the fermented soy yogurt results [32].

#### 3.1.4. Analysis of Volatile Flavour Substances in PPY

Most of the flavour compounds in PPY are associated with lactic acid metabolism, lipid peroxidation and degradation, protein hydrolysis, and amino acid metabolism [33]. The volatile flavour compounds in PPY after fermentation are presented in Table 4, with a total of 43 identified compounds, including 17 aldehydes, 10 alcohols, 4 acids, 9 ketones, 2 furans, and 1 ester. Among the aldehydes, some contributed to pleasant sensory characteristics such as fruity (citrus, banana, orange, and cherry), nutty (almond and pea), toasty, cinnamon, green, buttery, and creamy notes, while others imparted undesirable fatty and greasy aromas. Alcohols were associated with floral (lavender, rose, violet, and peony), fruity (banana, citrus, melon, grape, and tropical fruits), and alcoholic (whisky and red wine) notes, as well as fresh and clean aromas, though some were linked to mushroom-like, earthy, and pungent off-flavours. Acids played a role in both enhancing the aroma with cheesy and buttery notes and introducing sour, fatty, and rancid odours. Ketones contributed to fruity (pear, banana, and mandarin orange), wine-like, medicinal, caramel, and buttery aromas, but some were also responsible for pungent, musty, earthy, and mushroom-like smells. Furans exhibited both fruity and clean aromas, along with pungent and earthy notes, while the only ester identified had a refreshingly sweet and fruity aroma, enhancing the overall sensory complexity. The results indicate that fermentation significantly alters the volatile composition of PPY, enhancing its flavour complexity while reducing undesirable off-flavours, providing insights for improving the sensory quality of PPY alternatives.

#### 3.1.5. pH, Titratable Acidity, and Water-Holding Capacity

Acid formation directly affects the gelatinisation of yogurt, and the acid-producing capacity of lactic acid bacteria is an important indicator for evaluating the fermentation characteristics of strains, which can reflect the growth status and vitality of strains [34]. As shown in Figure 3, with the increase in fermentation time, the pH of PPY gradually decreased and the titratable acidity content increased. In the first 7 h of fermentation, the rate of pH decrease was high and the lactic acid bacteria grew normally. After 4 h, the PPY gradually formed a gel because the pH of the PPY all dropped below 6 one after another. Although the isoelectric point of pea protein has not been reached at pH 6, the acidic environment has begun to expose the hydrophobic regions hidden inside the molecule to exposure and hydrophobic interactions, causing the protein to begin pre-aggregation and micelle-like aggregation behaviour, similar to previous studies [35]. After 7 h, the pH value of each group of PPY decreased, but the pH value change was very small. At this stage, the content of nutrients required for the growth of lactobacilli decreased, the pH value of the PPY system was low, the growth of lactobacilli was inhibited, it was in the period of decline, and the capacity of acid production was weakened. Finally, at 10 h, the pH of PPY reached 4.66, close to the ideal value of 4.6, and the titratable acidity reached 84.96°T, which was in accordance with our standard ≥ 70°T.

The water-holding capacity of fermented milk is determined by the three-dimensional mesh structure of the proteins in fermented milk [36]. The strong water-holding capacity of fermented milk and the good organisation status can effectively prevent whey precipitation, and the gel stability is strong and, at the same time, makes the PPY’s texture softer and more delicate. From Figure 3B, it can be seen that the water-holding capacity gradually increased with the increase in fermentation time, and the water holding capacity increased greatly after 4 h, which was consistent with the time when the PPY formed the gel state.

#### 3.1.6. Textural Properties

Textural properties are an important part of PPY quality; it directly affects the taste and eating experience of PPY, through the determination of the texture meter, and can guide the development and production of PPY to obtain better-quality PPY products [37]. As shown in Table 5, the hardness and chewiness increased dramatically with an increase in fermentation time, which indicated that the gel of PPY had a strong resistance to deformation, and the gel structure was strong. The springiness and cohesiveness changed abruptly after 4 h, and the elasticity increased and approached 1, indicating that the protein network was gradually tightened and the recovery of the speed of deformation was accelerated. Additionally, the increase in the cohesiveness indicated that the internal connection was uniform, and it was not easy for the PPY to become deformed. The increase in cohesiveness indicated that the internal connection was uniform, and it was not easy to precipitate whey, while the change in adhesiveness was not significant, which was related to the formation of a stable gel network in PPY.

#### 3.1.7. Changes in Free Sulfhydryl Content During Fermentation

The free sulfhydryl group (-SH) is a critical functional group in protein molecules, contributing significantly to protein structural stability and functionality. During PPY fermentation, changes in free sulfhydryl content are closely linked to protein aggregation, cross-linking, and gel formation. Higher free sulfhydryl content typically correlates with increased protein aggregation and improved gel structure [21].

As PPY fermentation progresses, the pH gradually decreases, resulting in acidification. Under acidic conditions, free sulfhydryl groups within cysteine are released and subsequently oxidised into disulfide bonds. This oxidation process reduces free sulfhydryl group content (Figure 4A). However, the total sulphur-containing components remain constant, with reversible conversion occurring between disulfide bonds and free sulfhydryl groups. Consequently, a decrease in free sulfhydryl groups corresponds to an increase in disulfide bond content. Disulfide bonds play a vital role in stabilising the gel’s advanced structure and enhance key yoghurt quality attributes, including taste, texture, and overall stability.

#### 3.1.8. Changes in Protein Intermolecular Forces During Fermentation

The formation of protein gels predominantly relies on intermolecular interactions, such as ionic bonding, hydrogen bonding, hydrophobic interactions, and disulfide bonding [38]. These molecular forces in protein gels can be determined based on the solubility of gels in various denaturing agents [39]. Figure 4B illustrates the solubility of samples in different denaturing solvents. As depicted, solubility exhibited significant changes throughout the fermentation process, closely associated with fermentation duration. Initially, higher solubility in fractions S1 and (S2–S1), measuring 3.55 mg/mL and 1.12 mg/mL, respectively, and lower solubility in fractions (S3–S2) and (S4–S3), measuring 0.27 mg/mL and 0.15 mg/mL, respectively, indicated that ionic bonds, hydrogen bonds, and hydrophobic interactions maintained the natural structure of pea protein. With prolonged fermentation, the protein content in fractions S1 and (S2–S1) notably decreased, by 53.98% and 87.64%, respectively, at the end of fermentation. Conversely, the protein content in fractions (S3–S2) and (S4–S3) increased significantly, by 123.76% and 562.23%, respectively (*p* < 0.05). Following gel formation, contributions from intermolecular hydrogen and ionic bonds diminished, while contributions from hydrophobic interactions and disulfide bonds increased. These results demonstrate that hydrophobic interactions and disulfide bonds are the primary forces facilitating gel formation in PPY, differing from the findings of Liu et al. [40]. regarding soy yoghurt. The gradual reduction in solubility of fractions S1 and (S2–S1) during fermentation may be attributed to pea protein aggregation and binding processes occurring under fermentation conditions, wherein hydrogen and ionic bonds originally maintaining protein structure are disrupted, causing molecular structural changes.

#### 3.1.9. Changes in Rheological Properties During Fermentation

The storage modulus (G′) represents the PPY’s ability to recover its original shape, reflecting its elasticity, while the loss modulus (G″) represents the energy dissipated during deformation, indicative of the viscous component. Figure 4C illustrates the changes in G′ and G″ during the fermentation of PPY. As fermentation progressed, both the storage and loss moduli gradually increased, suggesting continuous improvement in PPY stability. Throughout fermentation, the storage modulus (G′) consistently exceeded the loss modulus (G″), indicating that elasticity contributed more significantly than viscosity and reflecting strong gel formation [41]. Figure 4B shows the ratio of loss modulus to storage modulus (G″/G′), which remained below 1 throughout fermentation, corroborating the findings from Figure 4D. This further supports the conclusion that the PPY exhibited desirable gelation properties, consistent with the findings reported by Tang et al. [42].

#### 3.1.10. SEM

SEM is a valuable tool for studying the microstructure of food products. As shown in Figure 5, PPY before fermentation (A) behaves as a liquid state, with more voids on the surface and cross-section, looser organisation, poor network structure, and no gel structure formed, whereas with the increasing fermentation time (B–D), the PPY gradually changes from a liquid state to a solid state, with the number of voids on the surface and cross-section gradually decreasing, the organisation gradually becoming more tightly packed, the aggregation of proteins occurring, and a dense and tightly latticed gel eventually forming a network.

### 3.2. The Effect of Different Treatments on the Flavour of PPY

Different treatments exhibit varying effects on the flavour profile of PPY. This study systematically compared the flavour characteristics of PPY produced using five different treatments to explore flavour differences and evaluate the effectiveness of bean flavour removal. Two physical methods were employed to mitigate the formation of bean flavour by reducing the activity of lipoxygenase (LOX), thereby decreasing the generation of lipid-derived volatile compounds. Additionally, a biological enzyme treatment was implemented by introducing alcohol dehydrogenase and aldehyde dehydrogenase, which catalyse the conversion of bean flavour compounds into other metabolites with minimal contribution to undesirable beany notes. Furthermore, a flavour-masking approach was investigated by incorporating coconut proteins, which interact with bean flavour functional groups to form new compounds that diminish the perception of beany off-flavours. Another masking strategy involved directly using coconut proteins to suppress the sensory perception of existing bean flavour compounds.

#### 3.2.1. Analysis and Comparison of Volatiles in PPY with Different Treatments

Figure 6 presents the Principal Component Analysis (PCA) score plot for samples A2–A6, providing a visual representation of the differences in flavour profiles among the PPY samples subjected to different treatments. In the PCA plot, a shorter distance between samples indicates a higher degree of similarity in their volatile compositions, whereas a greater distance signifies more pronounced differences. As shown in Figure 6, the flavour profiles of PPY samples subjected to different treatments exhibit noticeable variations. Additionally, the volatile compound compositions of A4, A5, and A6 display a certain degree of similarity, suggesting comparable effects of these treatments on the flavour characteristics of PPY.

Figure 7A presents the three-dimensional spectra of volatile organic compounds (VOCs) in PPY samples subjected to different treatments, with samples A2–A6 arranged sequentially from left to right. The figure provides a visual representation of the differences in VOC compositions among the various treatments. To facilitate observation, a top-down view of the spectra was used for further comparison.

As shown in Figure 7B, volatile flavour compounds are primarily concentrated within a migration time range of 1.0–1.5 on the horizontal axis and a retention time range of 100–400 on the vertical axis. To further visualise and compare the variations in volatile components among PPY samples treated with different methods, the fermented sample (A2) was selected as a reference. The spectra of the other samples were subtracted from this reference to generate a differential comparison plot, as shown in Figure 7C. In this plot, red regions indicate compounds present at higher concentrations compared to the reference sample, while blue regions represent compounds present at lower concentrations.

The results indicate that the microwave-treated sample (A3) exhibited the lowest levels of volatile flavour compounds compared to the other treatments. Additionally, the composition and concentration of volatile compounds in samples treated with A4, A5, and A6 showed a high degree of similarity, aligning with the findings of Principal Component Analysis (PCA), as previously described. The three treatments demonstrated a consistent trend in the increase and decrease in volatile compounds relative to A2. Notably, these treatments target different aspects of flavour modification: lipoxygenase inactivation, interactions with bean flavour functional groups, and mitigation of pre-existing bean flavour compounds. However, despite their differing mechanisms of action, their overall impact on flavour composition was similar. This suggests that a combination of these approaches could potentially yield more effective results, although further validation is required to confirm the synergistic effects.

#### 3.2.2. Comparative Analysis of Fingerprints of Volatile Components in PPY with Different Treatments

To provide a more detailed representation of the variation patterns and relative concentrations of volatile compounds across different samples under various treatment conditions, the Gallery Plot plug-in was utilised to generate fingerprint plots of the volatile substances. These fingerprint plots allow for a comprehensive visual comparison of the distribution and intensity of volatile compounds, facilitating the identification of differences in flavour profiles among the PPY samples subjected to different treatments.

To further elucidate the distribution and relative abundance of volatile compounds under different treatment conditions, Figure 8 highlights specific groups of compounds that exhibited the highest concentrations in different samples. Compounds enclosed within the red box were most abundant in the control (A2) and included (E, E)-2,4-hexadienal, furfural, cis-2-pentenol, and ethyl diphosgene, with cis-2-pentenol being the only compound associated with an unpleasant aroma. The yellow box indicates compounds that were highest in the microwave-treated sample (A3), including (E, E)-2,4-heptadienal, 3-methyl-2-butenal, butyric acid, 3-methylbutanol, ethyl acetate, 2-octanone, 2-pentanone, and 1-hexanol. While most of these compounds imparted fresh and fruity notes, certain volatile acids contributed irritatingly acidic and fatty off-flavours. The green box represents compounds that were most prevalent in the dual enzyme-treated sample (A5), including benzaldehyde, 3-octanol, 1-butanol, 1-heptanol, 2-hexanone, and 2-butanone, all of which contributed pleasant aromatic qualities. In contrast, compounds in the orange box reached their highest concentrations in the masking treatment (A6) and included octanal, glutaraldehyde, heptanal, and hexanal, all of which are associated with pungent and unpleasant flavours. The increased presence of off-flavours in this treatment suggests that the masking approach may be unsuitable for the fermentation of PPY.

However, previous research on flavour enhancement and sensory acceptance has indicated that the impact of volatile compounds on food flavour depends on their concentration and balance. While moderate levels of aroma-active compounds can enhance sensory appeal, excessive concentrations may lead to the development of undesirable off-flavours, thereby reducing consumer acceptance [42]. Therefore, achieving an optimal balance of key flavour compounds is crucial to improving the sensory attributes of PPY while avoiding the formation of off-flavours.

#### 3.2.3. Effectiveness of Different Treatments on the Removal of Bean Flavour from PPY

The presence of bean flavour is a common characteristic of legumes and other plant-based proteins, often posing a major challenge to consumer acceptance of plant-based food products. As demonstrated in the findings above, the five different treatments influenced the flavour profile of PPY to varying degrees. However, a more detailed analysis is required to precisely evaluate their effectiveness in reducing bean flavour. Further investigation should focus on quantifying the key volatile compounds associated with bean flavour and assessing the sensory perception of these changes to determine the most effective strategy for improving the flavour acceptability of PPY.

Hexanal, 2-pentylfuran, 1-octen-3-ol, and pyrazines are well-recognised as key flavour compounds in bean-based products, all of which possess low sensory thresholds and significantly impact the overall flavour profile [43]. In this study, pyrazines were not detected, likely due to the use of pea protein powder, which undergoes compositional changes in volatile compounds following protein extraction. Among these compounds, hexanal is closely associated with bean-like and grassy aromas in bean milk [44]. A comparison with the control sample (A2) (Figure 9C) revealed that hexanal levels increased in all four treated samples, with high-temperature and high-pressure treatment (A4) yielding hexanal concentrations comparable to the control. However, fermentation led to a substantial reduction in hexanal levels compared to pre-fermentation levels (Figure 9A), likely due to lipoxygenase (LOX) inactivation by microbial fermentation, thereby suppressing the enzymatic oxidation of lipids responsible for generating bean-flavour compounds [45]. 2-Pentylfuran and 1-octen-3-ol are primarily produced through non-enzymatic oxidation, a key pathway in bean flavour formation during bean milk processing [46]. This oxidation process is driven by the presence of unsaturated fatty acids such as linoleic acid, linolenic acid, and arachidonic acid, which are highly susceptible to reactive oxygen species (ROS). Due to their unsaturated double bonds, these fatty acids undergo hydrogen abstraction initiated by various free radicals, leading to the formation of hydroperoxides (HPODs) through a free radical chain reaction [31]. 2-Pentylfuran, characterised by grassy and earthy notes, is produced via the oxidation of linoleic acid by singlet oxygen [47]. The study results indicate that 2-pentylfuran levels increased in all four treatment groups relative to A2, with microwave treatment (A3) exhibiting concentrations comparable to the control (A2). However, fermentation resulted in a roughly 50% reduction in 2-pentylfuran content, though its decrease was less pronounced than that of hexanal. 1-Octen-3-ol, known for its mushroom-like aroma, exhibited a decline in all treatment groups except for A3, where its concentration remained similar to that of A2. Notably, fermentation had little impact on the levels of 1-octen-3-ol, suggesting that microbial activity plays a limited role in altering its presence in PPY. These findings highlight the differential effects of fermentation and processing treatments on key flavour compounds, with microbial fermentation significantly reducing hexanal and 2-pentylfuran but exerting minimal influence on 1-octen-3-ol.

Lv et al. [48] employed dynamic headspace dilution method-olfaction-mass spectrometry (DHDA-GC-O-MS) to identify eight key volatile compounds influencing the odour of peas, including aldehydes (benzaldehydes and (E, E)-2,4-decadienal), alcohols (1-octen-3-ol, (E)-hexenal, alditol, and pentanol), and acids (acetic acid). In the present study, only hexanal and 1-octen-3-ol were detected in PPY samples, which is consistent with the findings reported by Lu et al. Similarly, VARA et al. [49] identified 3-methyl-1-butanol, pentanol, 1-octen-3-ol, trans, trans-2,4-heptadienal, acetophenone, 1-octen-3-one, and 3-isopropyl-2-methoxypyrazine as the primary contributors to pea odour. In this study, 3-methylbutanol, 1-pentanol, and (E, E)-2,4-heptadienal were detected, partially aligning with the previous research. Notably, compared to the control (A2), all four treatment groups exhibited a significant reduction in (E, E)-2,4-heptadienal, suggesting that these processing methods effectively mitigated this pea odour-related compound. In contrast, 3-methylbutanol and 1-pentanol followed a similar trend in concentration changes across different treatments. Microwave treatment (A3) and dual enzyme treatment (A5) led to an increase in both compounds, whereas autoclave and high-pressure treatments (A4) and the masking method (A6) resulted in a decrease in their concentrations. These variations suggest that different processing methods influence the formation and degradation pathways of pea odour compounds, which could have implications for optimising flavour profiles in PPY.

Hexanal is a major component in bean volatiles; however, it does not solely contribute to bean odour. Instead, when combined with other bean and non-bean volatile compounds, it results in the characteristic bean odour profile. The perception of bean odour is influenced by a combination of volatile compounds with multiple organoleptic attributes, including musty/dusty, mildewy/earthy, as well as green/pea-pod, nutty, and brown notes. As a result, grassy and earthy aromas are key components of pea flavour [12]. In this study, ten compounds associated with grassy and earthy flavours were identified in the samples, including (E)-2-Heptenal, (E)-2-Hexenal, 1-Pentanal, (E, E)-2,4-Hexadienal, 1-Hexanol, (Z)-2-Pentenol, 1-Penten-3-ol, 3-Octanol, Cyclohexanone, and 2-Pentylfuran. A comparison with A2 revealed that different treatments had varying degrees of impact on reductions in these compounds. Specifically, microwave treatment (A3) reduced the content of four compounds, autoclave treatment (A4) reduced seven compounds, dual-enzyme treatment (A5) reduced six compounds, and masking treatment (A6) reduced eight compounds. Among the treatments, high-temperature and high-pressure treatment (A4) emerged as the most effective method for reducing bean flavour in PPY, as it significantly decreased the content of undesirable volatiles. In contrast, dual-enzyme treatment (A5) resulted in excessive acetaldehyde, which may negatively impact the overall flavour balance. Additionally, masking treatment (A6) exhibited the highest concentrations of several unpleasant volatile compounds, suggesting that it may not be the optimal approach for flavour improvement. These findings highlight the importance of selecting appropriate processing methods to enhance the sensory properties of PPY while minimising undesirable bean flavour attributes.

## 4. Conclusions

Fermentation is an effective way to improve the physicochemical properties, structure, nutritional quality, and degradation of anti-nutritional factors of PPY. The PPY was fermented according to the optimal fermentation conditions obtained, and the observation of the changes in physicochemical properties, nutritional properties, water-holding properties, textural properties, microstructure, rheological properties, as well as free sulfhydryl groups and intermolecular forces during the fermentation process revealed that as the fermentation time proceeded, the pH decreased, proteins were aggregated, and the gel state was gradually formed, while the protein content was increased. Comparison of before and after fermentation revealed that the content of trypsin inhibitors decreased about three times after fermentation, and the effect of degrading anti-nutritional factors was extremely significant. Hydrophobic forces and disulfide bonds were the main forces involved in the formation of the gel state in PPY. Furthermore, fermented PPY exhibited richer and more pleasant flavours, with substantially reduced levels of characteristic pea-related off-flavours such as hexanal and 2-pentylfuran.

The presence of inherent pea off-flavours significantly restricts the market acceptance of PPY, and existing deodorisation methods often lack effectiveness. This study confirmed that fermentation significantly enhances the sensory appeal by increasing desirable flavours (e.g., esters and ketones) and reducing undesirable pea-flavour compounds. Among the five treatments evaluated, high temperature and pressure (A4) achieved the most effective deodorisation, whereas microwave treatment (A3) demonstrated the lowest efficiency due to uneven protein denaturation. High-temperature and pressure treatment (A4), dual-enzyme hydrolysis (A5), and flavour-masking (A6), although targeting different mechanisms (enzyme inactivation, hydrolysis, and encapsulation), effectively reduced pea off-flavours, resulting in convergent flavour profiles. Principal component analysis (PCA) and two-dimensional mapping further supported a potential synergistic effect, suggesting that combining these treatments could yield enhanced deodorisation results. This outcome aligns with broader research on plant proteins, such as soybean, where thermal and enzymatic methods are also key to mitigating beany flavours by targeting lipid oxidation products. These methods can be applied to other plant proteins as well.

This research provides a robust theoretical foundation for the development and flavour enhancement of PPY. Its fermentation mechanism and technological framework can also be extended to other plant-based dairy alternatives, promoting sustainable food system development. Future studies could further explore the nutritional functionalities of PPY; develop diverse functional PPY products; integrate high-temperature, pressure, and dual-enzyme methods with physical adsorption (masking method) to construct a multi-stage deodorisation system; and utilise metabolomics to elucidate the metabolic pathways of pea flavour precursors. Investigating key enzyme targets (e.g., lipid oxidases) through metabolomics would further clarify deodorisation mechanisms and enhance understanding of flavour transformation processes.

## Figures and Tables

**Figure 1 foods-14-03363-f001:**
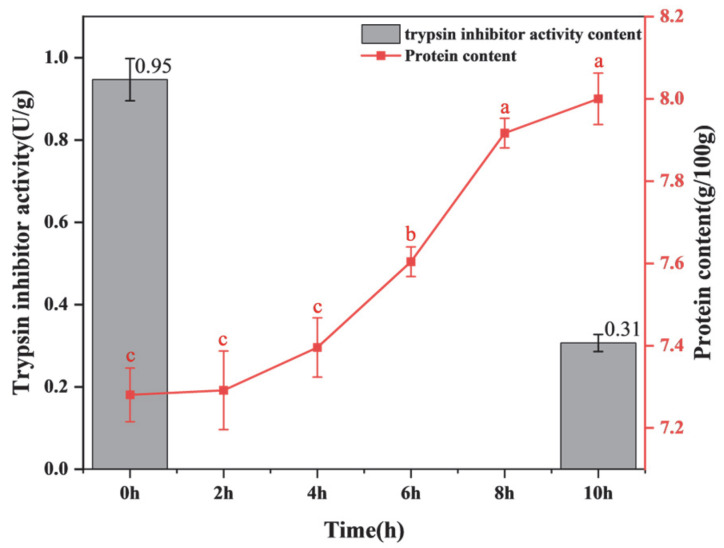
Changes in protein content and trypsin inhibitor activity during fermentation. a–c: A way of labeling the results of significance analysis. Identical letters indicate that there is no significant difference between the two data sets (*p* > 0.05). Completely different letters indicate a significant difference between the two data sets (*p* < 0.05).

**Figure 2 foods-14-03363-f002:**
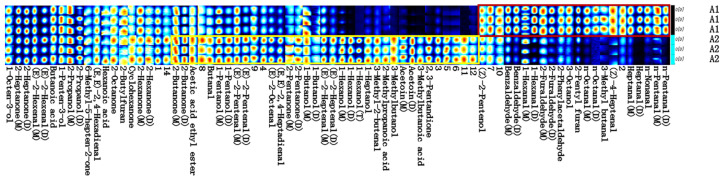
Fingerprints of volatiles in PPY before and after fermentation. m, monomer; d, dimer. Each row of the graph represents the volatile content of the samples, while each column represents the signal peaks of volatile substances in different samples. The brightness and darkness of the signal peaks indicate the concentration of the substance. The brighter the colour, the higher the volatile content, while the darker the colour, the lower the volatile content.

**Figure 3 foods-14-03363-f003:**
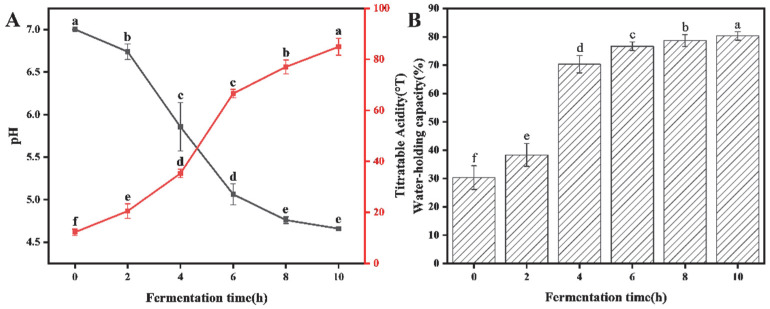
Changes in pH, titratable acidity, and water-holding capacity during fermentation. (**A**) Changes in pH and titratable acidity during fermentation; (**B**) Changes in water holding capacity during fermentation. a–f: A way of labeling the results of significance analysis. Identical letters indicate that there is no significant difference between the two data sets (*p* > 0.05). Completely different letters indicate a significant difference between the two data sets (*p* < 0.05).

**Figure 4 foods-14-03363-f004:**
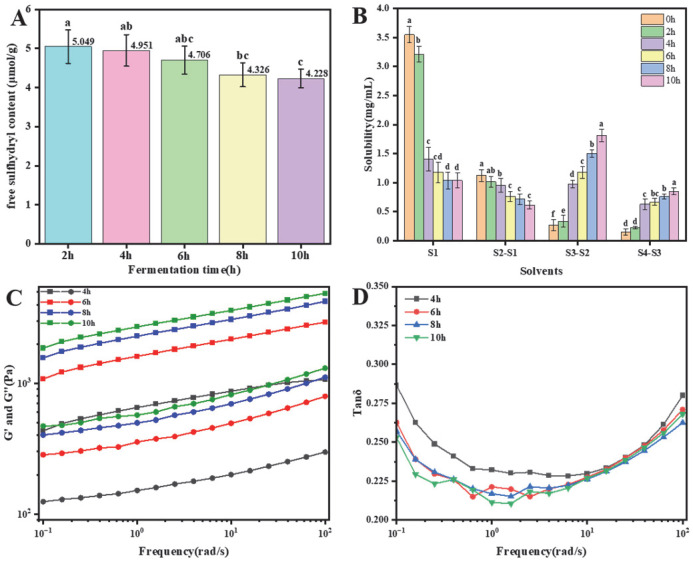
Fermentation mechanism study of PPY. (**A**) Changes in free sulfhydryl content during fermentation. (**B**) Changes in intermolecular forces during fermentation. (**C**,**D**) Changes in rheological properties during fermentation. a–f: A way of labeling the results of significance analysis. Identical letters indicate that there is no significant difference between the two data sets (*p* > 0.05). Completely different letters indicate a significant difference between the two data sets (*p* < 0.05).

**Figure 5 foods-14-03363-f005:**
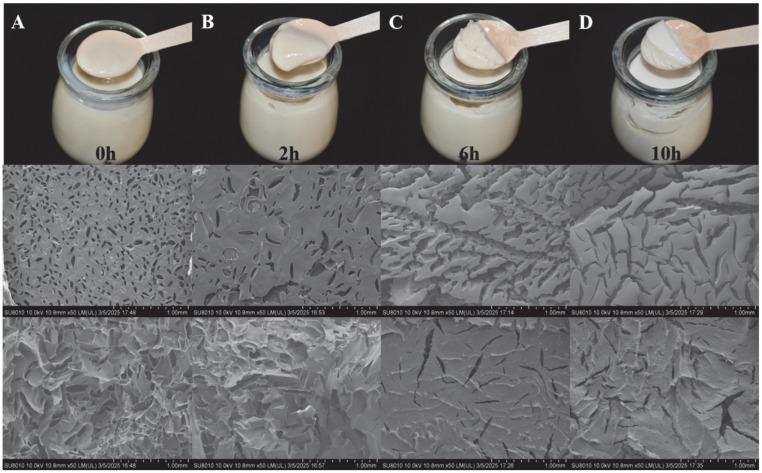
State and microstructure during fermentation of PPY. (**A**–**D**): This column shows the state diagrams, surface microstructure, and cross-section microstructure of PPY at 0 h, 2 h, 6 h, and 10 h of fermentation, respectively.

**Figure 6 foods-14-03363-f006:**
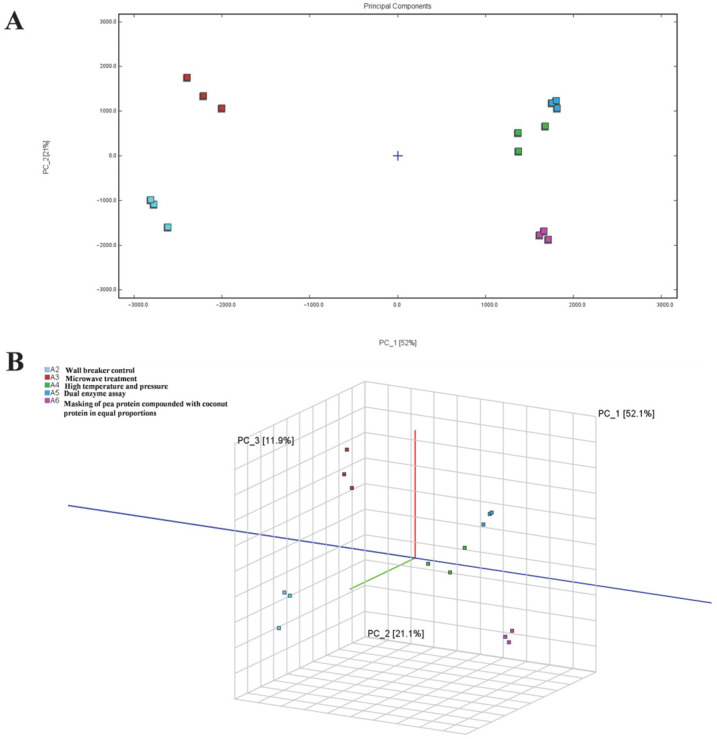
Plot of PCA scores of volatile components in PPY with different treatments. A2, wall breaker control; A3, microwave treatment; A4, high temperature and pressure; A5, dual enzyme assay; A6, masking of pea protein compounded with coconut protein in equal proportions. (**A**) 2D Scatterplot; (**B**) 3D Scatterplot.

**Figure 7 foods-14-03363-f007:**
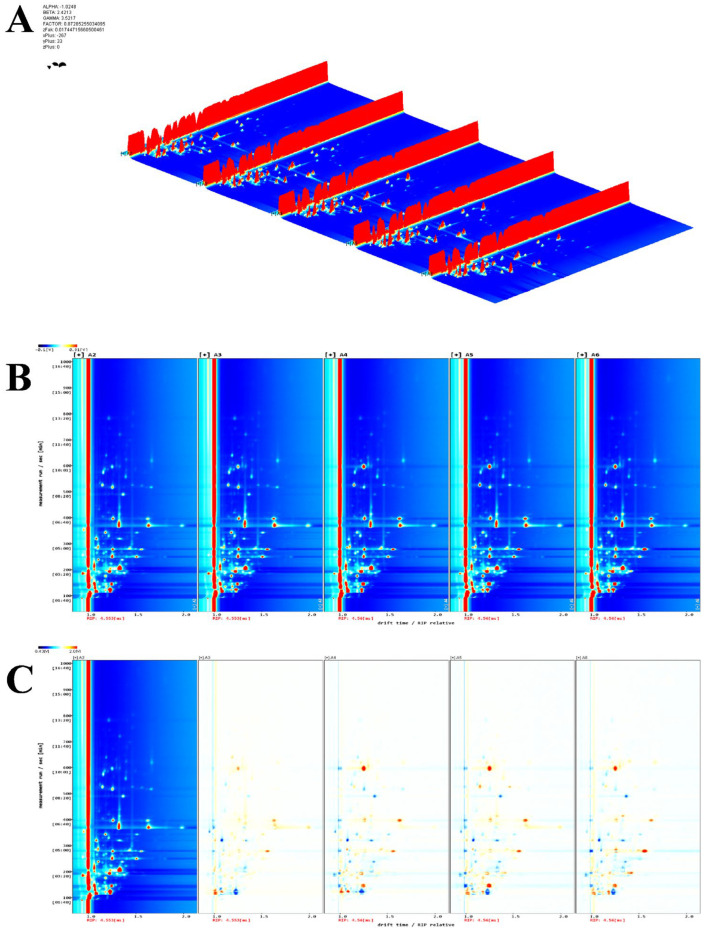
GC-IMS 3D and 2D spectra of volatiles in PPY with different treatments. A2, wall breaker control; A3, microwave treatment; A4, high temperature and pressure; A5, dual enzyme assay; A6, masked by equal ratio of pea protein and coconut protein compound. a: three-dimensional spectra; b: two-dimensional spectra; c: two-dimensional difference spectra. (**A**) three-dimensional spectrum. The volatile components in PPY with different treatments were analysed by gas-ion mobility spectrometry (GC-IMS), and three-dimensional spectra of volatiles were obtained (Figure 7A), where the *x*-axis indicates the ion migration time, the *y*-axis the retention time, and the *z*-axis the signal intensity. The unfermented (A1) and fermented (A2) samples are arranged in order from left to right. (**B**) two-dimensional spectrum; (**C**) two-dimensional difference spectrum. Each bright spot in the spectrum represents a volatile compound, but some analytes with high proton affinity, such as monomers, dimers, and other different forms, may have multiple spots. The background of the 2D top view is blue, while the red vertical line at 1.0 in the horizontal coordinate is the RIP peak (reactive ion peak, normalised). Depending on the presence or absence of peaks (colour dots) or colour depths, it is possible to visually represent differences in composition and concentration between samples. The vertical coordinate indicates the measurement run time of the gas chromatograph and the horizontal coordinate indicates the ion migration time (normalised.) Each point on either side of the RIP peak represents a volatile organic compound. The colour represents the concentration of the substance, with white indicating a lower concentration; red indicating a higher concentration; and the darker the colour, the higher the concentration.

**Figure 8 foods-14-03363-f008:**
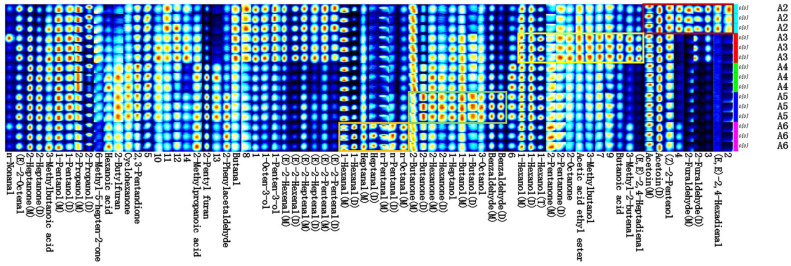
Fingerprints of volatiles in PPY with different treatments. M, monomer; D, dimer.

**Figure 9 foods-14-03363-f009:**
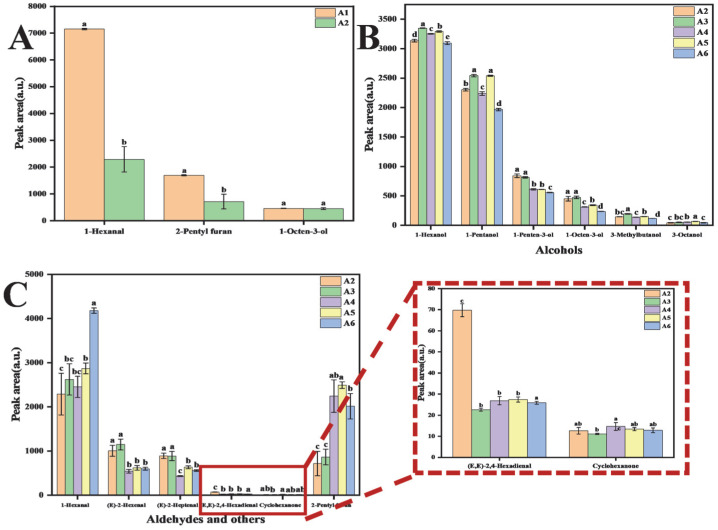
Comparison of peak areas of sourdough bean odour obtained by different treatments. (**A**) Comparison of three flavor substances before and after fermentation; (**B**) Comparison of flavor substances obtained by different treatments; (**C**) Comparison of flavor substances obtained by different treatments. a–e: A way of labeling the results of significance analysis. Identical letters indicate that there is no significant difference between the two data sets (*p* > 0.05). Completely different letters indicate a significant difference between the two data sets (*p* < 0.05).

**Table 1 foods-14-03363-t001:** Amount of each solution added for trypsin solution activity determination.

Addition	Samples (mL)	Standard (mL)
L-BAPA solution	5	5
Tris-CaCl_2_ solution	1	2
Trypsin solution	2	2
Sample Diluent	1	0
Acetate solution	1	1

**Table 2 foods-14-03363-t002:** Information table of samples treated by different methods.

Categories	Number	Methods	Terms
Physical method	A2	Wall breaker heating	Bean milk mode
	A3	Microwave heating	Power: 500 w; time: 5 min
	A4	High temperature, high voltage	Temperature: 121 °C; time: 3 min
Bioenzymatic method	A5	Double enzymatic method	Alcohol dehydrogenase: 0.5%; aldehyde dehydrogenase: 0.5%; 2 h
Hidden method	A6	Complexed with coconut protein	Compounding ratio: 1:1

**Table 3 foods-14-03363-t003:** Gas chromatographic conditions.

Time (min)	E1 (mL/min)	E2 (mL/min)	R
0	75.0	2.0	recording
2	75.0	2.0	-
10	75.0	10.0	-
20	75.0	100.0	-
30	75.0	150.0	stop

**Table 4 foods-14-03363-t004:** Breakdown of volatile components in PPY before and after fermentation.

Number	Compound	CAS	Formula	MW	RI	Rt (s)	Dt (RIPrel.)	Flavour Description
	Aldehyde							
1	n-Nonanal	C124196	C_9_H_18_O	142.2	1109.5	803.208	1.47121	rose, citrus, strong oily
2	(E)-2-Octenal	C2548870	C_8_H_14_O	126.2	1070.4	720.012	1.33691	fresh cucumber, fatty, green herbal, banana, green leaf
3	(E, E)-2,4-Heptadienal	C4313035	C_7_H_10_O	110.2	1028.5	640.37	1.19347	fatty, oily, aldehyde, vegetable, cinnamon
4	n-Octanal(M)	C124130	C_8_H_16_O	128.2	1019.5	624.442	1.40351	aldehyde, waxy, citrus, orange, fruity, fatty
5	n-Octanal(D)	C124130	C_8_H_16_O	128.2	1019.8	624.91	1.82102	aldehyde, waxy, citrus, orange, fruity, fatty
6	Benzaldehyde(M)	C100527	C_7_H_6_O	106.1	969.2	526.452	1.15271	bitter almond, cherry, nutty
7	Benzaldehyde(D)	C100527	C_7_H_6_O	106.1	968.2	524.411	1.4752	bitter almond, cherry, nutty
8	(E)-2-Heptenal(M)	C18829555	C_7_H_12_O	112.2	964.9	517.706	1.25648	spicy, green vegetables, fresh, fatty
9	(E)-2-Heptenal(D)	C18829555	C_7_H_12_O	112.2	964.7	517.414	1.67382	spicy, green vegetables, fresh, fatty
10	Heptanal(M)	C111717	C_7_H_14_O	114.2	907.0	413.271	1.33583	fresh, aldehyde, fatty, green herbs, wine, fruity
11	Heptanal(D)	C111717	C_7_H_14_O	114.2	907.4	414.05	1.70265	fresh, aldehyde, fatty, green herbs, wine, fruity
12	(E)-2-Hexenal(M)	C6728263	C_6_H_10_O	98.1	855.9	344.24	1.1882	green, banana, fat
13	(E)-2-Hexenal(D)	C6728263	C_6_H_10_O	98.1	855.7	343.98	1.52152	green, banana, fat
14	2-Furaldehyde(M)	C98011	C_5_H_4_O_2_	96.1	834.5	319.688	1.08484	sweet, woody, almond, bready
15	2-Furaldehyde(D)	C98011	C_5_H_4_O_2_	96.1	833.7	318.831	1.33887	sweet, woody, almond, bready
16	1-Hexanal(M)	C66251	C_6_H_12_O	100.2	797.0	280.864	1.25457	fresh, green, fat, fruity
17	1-Hexanal(D)	C66251	C_6_H_12_O	100.2	794.0	278.009	1.56961	fresh, green, fat, fruity
18	3-Methyl-2-butenal	C107868	C_5_H_8_O	84.1	781.3	265.725	1.09303	fruity
19	(E)-2-Pentenal(M)	C1576870	C_5_H_8_O	84.1	751.5	237.255	1.1097	potato, peas
20	(E)-2-Pentenal(D)	C1576870	C_5_H_8_O	84.1	751.7	237.421	1.363	potato, peas
21	n-Pentanal(M)	C110623	C_5_H_10_O	86.1	698.5	193.887	1.19182	green grassy, faint banana, pungent
22	n-Pentanal(D)	C110623	C_5_H_10_O	86.1	699.2	194.386	1.42632	green grassy, faint banana, pungent
23	Butanal	C123728	C_4_H_8_O	72.1	607.8	150.243	1.29702	pungent, fruity, green leaf
24	2-Phenylacetaldehyde	C122781	C_8_H_8_O	120.2	1044.9	670.509	1.25626	hyacinth, sweet fruity, almond, cherry, clover honey, cocoa
25	(Z)-4-Heptenal	C6728310	C_7_H_12_O	112.2	906.0	411.73	1.14901	grass, oil
26	(E, E)-2,4-Hexadienal	C142836	C_6_H_8_O	96.1	919.0	433.066	1.11539	sweet, green, floral, citrus
27	3-Methyl butanal	C590863	C_5_H_10_O	86.1	661.9	173.769	1.40451	chocolate, fat
	Alcohol							
1	1-Octen-3-ol	C3391864	C_8_H_16_O	128.2	991.5	574.264	1.16386	mushroom, lavender, rose, hay
2	1-Heptanol	C111706	C_7_H_16_O	116.2	983.8	557.355	1.39708	grape, fruity, wine, violet, peony
3	1-Hexanol(M)	C111273	C_6_H_14_O	102.2	877.2	370.451	1.32214	fresh, fruity, wine, sweet, green
4	1-Hexanol(D)	C111273	C_6_H_14_O	102.2	876.9	370.191	1.63872	fresh, fruity, wine, sweet, green
5	1-Hexanol(T)	C111273	C_6_H_14_O	102.2	877.2	370.451	2.00249	fresh, fruity, wine, sweet, green
6	(Z)-2-Pentenol	C1576950	C_5_H_10_O	86.1	771.1	255.667	0.94181	green, plastic, rubber
7	1-Pentanol(M)	C71410	C_5_H_12_O	88.1	764.1	248.884	1.25551	balsamic
8	1-Pentanol(D)	C71410	C_5_H_12_O	88.1	765.0	249.819	1.51474	balsamic
9	3-Methylbutanol	C123513	C_5_H_12_O	88.1	734.0	221.968	1.24723	whiskey, banana, fruity
10	1-Penten-3-ol	C616251	C_5_H_10_O	86.1	688.3	186.576	0.94446	ethereal, green, tropical fruity
11	2-Propanol(M)	C67630	C_3_H_8_O	60.1	519.8	116.007	1.09451	alcohol, spicy
12	2-Propanol(D)	C67630	C_3_H_8_O	60.1	516.0	114.699	1.23601	alcohol, spicy
13	3-Octanol	C589980	C_8_H_18_O	130.2	994.8	581.591	1.40248	earth, mushrooms, herb, melon, citrus, woody
14	1-Butanol(M)	C71363	C_4_H_10_O	74.1	665.6	175.511	1.18867	wine
15	1-Butanol(D)	C71363	C_4_H_10_O	74.1	667.0	176.196	1.38267	wine
	Acids							
1	3-Methylbutanoic acid	C503742	C_5_H_10_O_2_	102.1	844.9	331.392	1.21796	sour, foot sweat, cheese
2	Butanoic acid	C107926	C_4_H_8_O_2_	88.1	816.9	300.847	1.16027	strong acetic acid, cheese, butter, fruity
3	Hexanoic acid	C142621	C_6_H_12_O_2_	116.2	1005.3	600.109	1.30445	sour, fatty, cheese, pungent, Daqu liquor
4	2-Methylpropanoic acid	C79312	C_4_H_8_O_2_	88.1	778.9	263.332	1.16816	yogurt, rancid cream
	Ketone							
1	2-Heptanone(M)	C110430	C_7_H_14_O	114.2	896.4	396.662	1.26582	pear, banana, fruity, slight medicinal fragrance
2	2-Heptanone(D)	C110430	C_7_H_14_O	114.2	896.6	396.922	1.62807	pear, banana, fruity, slight medicinal fragrance
3	2-Pentanone(M)	C107879	C_5_H_10_O	86.1	690.3	187.906	1.1186	acetone, fresh, sweet fruity, wine
4	2-Pentanone(D)	C107879	C_5_H_10_O	86.1	690.9	188.404	1.37289	acetone, fresh, sweet fruity, wine
5	2-Butanone(M)	C78933	C_4_H_8_O	72.1	591.3	143.701	1.06993	fruity, camphor
6	2-Butanone(D)	C78933	C_4_H_8_O	72.1	595.2	145.228	1.24957	fruity, camphor
7	6-Methyl-5-hepten-2-one	C110930	C_8_H_14_O	126.2	998.0	587.988	1.17827	citrus, fruity, mouldy, ketone
8	2-Octanone	C111137	C_8_H_16_O	128.2	1002.8	595.975	1.33448	mouldy, ketone, milk, cheese, mushroom
9	Cyclohexanone	C108941	C_6_H_10_O	98.1	901.1	404.043	1.15826	strong pungent, earthy
10	2-Hexanone(M)	C591786	C_6_H_12_O	100.2	784.6	269.111	1.19074	fruity, fungal, meaty, buttery
11	2-Hexanone(D)	C591786	C_6_H_12_O	100.2	783.5	268.002	1.5009	fruity, fungal, meaty, buttery
12	2,3-Pentandione	C600146	C_5_H_8_O_2_	100.1	696.2	192.175	1.2356	sweet, cream, caramel, nuts, cheese
13	Acetoin(M)	C513860	C_4_H_8_O_2_	88.1	713.4	205.186	1.07012	butter, cream
14	Acetoin(D)	C513860	C_4_H_8_O_2_	88.1	712.5	204.521	1.33232	butter, cream
	Others							
1	2-Pentyl furan	C3777693	C_9_H_14_O	138.2	1003.8	597.587	1.25537	bean, fruity, earthy, green, vegetable
2	2-Butylfuran	C4466244	C_8_H_12_O	124.2	897.8	398.744	1.17971	mild fruity, alcoholic, sweet and spicy
3	Acetic acid ethyl ester	C141786	C_4_H_8_O_2_	88.1	616.9	153.95	1.10298	fresh, fruity, sweet, grassy

**Table 5 foods-14-03363-t005:** Changes in textural properties during fermentation.

Fermentation Time	Hardness (g)	Adhesiveness (mJ)	Cohesiveness	Springiness (%)	Chewiness (mJ)
0 h	5.13 ± 0.35 ^e^	0.07 ± 0.01 ^bc^	0.80 ± 0.33 ^a^	−21.67 ± 16.17 ^c^	−0.03 ± 0.02 ^e^
2 h	5.90 ± 0.26 ^e^	0.11 ± 0.03 ^a^	0.34 ± 0.29 ^b^	9.00 ± 6.56 ^b^	0.00 ± 0.01 ^e^
4 h	17.73 ± 0.60 ^d^	0.09 ± 0.01 ^ab^	0.89 ± 0.03 ^a^	93.67 ± 3.06 ^a^	0.44 ± 0.04 ^d^
6 h	95.13 ± 7.46 ^c^	0.04 ± 0.02 ^cd^	0.90 ± 0.03 ^a^	95.67 ± 0.58 ^a^	2.41 ± 0.18 ^c^
8 h	129.93 ± 9.02 ^b^	0.03 ± 0.01 ^d^	0.85 ± 0.02 ^a^	95.00 ± 1.73 ^a^	3.08 ± 0.21 ^b^
10 h	153.60 ± 7.58 ^a^	0.04 ± 0.01 ^cd^	0.82 ± 0.04 ^a^	93.33 ± 3.22 ^a^	3.45 ± 0.40 ^a^

^a–e^: A way of labeling the results of significance analysis. Identical letters indicate that there is no significant difference between the two data sets (*p* > 0.05). Completely different letters indicate a significant difference between the two data sets (*p* < 0.05).

## Data Availability

The original contributions presented in the study are included in the article. Further inquiries can be directed to the corresponding author.

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
