# Peer review of "The Fermentation Mechanism of Pea Protein Yogurt and Its Bean Odour Removal Method"

_foods, 2025, doi:10.3390/foods14193363_

Round 1
Reviewer 1 Report
Comments and Suggestions for Authors
The topic entitled (The Fermentation Mechanism of Pea Protein Yogurt and Its Bean Odor Removal Method) was reviewed.
General comments
- The authors presented pea protein yoghurt as the main subject. However, there are several terms used in the manuscript to indicate this product such as pea protein yoghurt, pea protein yogurt, yogurt, and “plant-based yogurt. Please use a consistent term to avoid confusion.
- I was confused on section 2.4. The authors developed pea protein yoghurt or soy yoghurt?
- There is lack of citations in several key findings presented in the manuscript.
- The authors should expand the discussion on the important phenomenon of each experiment.
Specific comments
- Section 2.1; Bacterial names should be italicized.
- Section 2.3
- The authors mentioned using of fructose in graphical abstract whereas mention sucrose in section 2.3.
- Was 10% of pea protein resulting in very thick texture of yogurt? The authors should clarify on the type of yoghurt developed in the current study.
- Please specify that water used in sample preparation was sterile.
- What was the conditions for pasteurization.
- The authors did not mention on the method of mother culture cultivation.
- The optimal temperature to culture yogurt starter typically ranges at 40-43ºC.
- Please provide more detail in section 2.12.1.
- Provide full name of IMS in section 2.12.5.
- Mention to the table or figure represented the result of section 3.1.
- Section 3.1.1; Discuss more on how protein content could increase during fermentation.
- Figure 1 did not show protein concentration at 0 h.
- Section 3.1.2; Discuss more on how fermentation could reduce trypsin inhibitor.
- Section 3.1.4; Clarify the role of pH 6 during yogurt fermentation.
- There is no section 3.1.5.
- Section 3.1.7; Mention to amino acid contained in pea protein which could form disulfide-bond.
Author Response
Comments 1: The authors presented pea protein yoghurt as the main subject. However, there are several terms used in the manuscript to indicate this product such as pea protein yoghurt, pea protein yogurt, yogurt, and “plant-based yogurt. Please use a consistent term to avoid confusion.
Response 1: Thank you for pointing this out. We agree with this comment. Therefore, we standardized the terminology-pea protein yogurt (PPY).
Comments 2: I was confused on section 2.4. The authors developed pea protein yoghurt or soy yoghurt?
Response 2: Thank you for pointing this out. This should have been Pea Protein Yogurt and I apologize for such a cheap mistake. We have corrected it in the text.
Line 126-128: Evaluation of water holding capacity (WHC) of PPY: the water holding capacity of PPY was measured according to the method of Delikanli et al.
Comments 3: There is lack of citations in several key findings presented in the manuscript.
Response 3: Thank you for pointing this out. We agree with this comment. Therefore, we have added references where appropriate.
Comments 4: The authors should expand the discussion on the important phenomenon of each experiment.
Response 4: Thank you for pointing this out. We agree with this comment. Therefore, we have added the discussion in the appropriate place.
Comments 5: Section 2.1; Bacterial names should be italicized.
Response 5: Thank you for pointing this out. We agree with this comment. Therefore, we've made the change in the article.
Line 94-96: Pea protein powder, purchased from Yantai Shuangta Foodstuffs Co. Ltd; leavening agent was a mixed compound of Lactobacillus bulgaricus and Streptococcus thermophilus, purchased from Yiran Biological Co. Ltd. of Shijiazhuang City, Hebei Province, China;
Comments 6: Section 2.3; The authors mentioned using of fructose in graphical abstract whereas mention sucrose in section 2.3.
Response 6: Thank you for pointing this out. We are using fructose, which has been corrected in the text. I apologize for this error in the translation where there was some confusion of words.
Line 113-115: Pea protein powder was mixed with water at a material-liquid ratio of 1:10, 5% fructose was added, high-pressure homogenization was carried out, and soymilk was made using a wall-breaker (BM9, Shenzhen, China), a process that included a heating step.
Comments 7: Section 2.3; Was 10% of pea protein resulting in very thick texture of yogurt? The authors should clarify on the type of yoghurt developed in the current study.
Response 7: Thank you for pointing this out. It is true that the 10% pea protein texture is thick, but in our study we wanted to make its flavor more prominent thus facilitating subsequent improvements to the flavor. In addition this yogurt we made is a solid yogurt and the high protein content increases its gel strength. In addition, we have already elaborated on yogurt types in the section 2.3.
Line 116-121: Then the soymilk was separated from other solid residues using an 80-mesh strainer to obtain the PPY pre-fermentation solution A1, and then 0.006% of the fermenter bacterial powder was added for fermentation after being cooled down, and fermentation was carried out for 10 h at a temperature of 37℃, and then post-cooking was carried out for 24 h at a temperature of 4℃ to obtain the PPY A2. After the end of the storage in 4 ℃ refrigerator, is a low temperature live bacteria type of solid yogurt.
Comments 8: Section 2.3; Please specify that water used in sample preparation was sterile.
Response 8: Thank you for pointing this out. We agree with this comment. We have stated in the text that the water is sterile.
Line 113-116: Pea protein powder was mixed with sterile water at a material-liquid ratio of 1:10, 5% fructose was added, high-pressure homogenization was carried out, and soymilk was made using a wall-breaker (BM9, Shenzhen, China), a process that included a heating step.
Comments 9: Section 2.3; What was the conditions for pasteurization.
Response 9: Thank you for pointing this out. We chose to heat and boil the raw pea protein powder directly during the heating process to inhibit anti-nutritional factors, and stored it directly in a 4°C refrigerator at the end of fermentation, with no pasteurization involved.
Comments 10: Section 2.3; The authors did not mention on the method of mother culture cultivation.
Response 10: Thank you for pointing this out. Our ferments are commercial ferments purchased directly from us and do not require culture activation and are weighed directly at the time of inoculation.
Comments 11: Section 2.3; The optimal temperature to culture yogurt starter typically ranges at 40-43°C.
Response 11: Thank you for pointing this out. We agree with this comment. It's true that yogurt ferments usually ferment at 40-43°C, but the optimal fermentation temperature offered by the merchant of this commercial ferment we chose is 37°C.
Comments 12: Please provide more detail in section 2.12.1.
Response 12: Thank you for pointing this out. We agree with this comment. Therefore, we have reorganized the information in more detail to explain the treatment here.
Line 223-229: As shown in Table 2, the preparation process of A2 was consistent with that of Section 2.3; A3 changed the heating of the wall-breaker in Section 2.3 to microwave heating (500w, 5min); A4 changed the heating of the wall-breaker in Section 2.3 to autoclaving (121°C, 3min); A5 was based on the enzyme digestion of A2 prior to heating in the wall-breaker (Alcohol dehydrogenase: 0.5%, aldehyde dehydrogenase: 0.5%, 2h); A6 was to replace the ingredients with a 1:1 blend of pea protein powder and coconut protein powder, and the other treatments were the same as A2.
Comments 13: Provide full name of IMS in section 2.12.5.
Response 13: Thank you for pointing this out. We agree with this comment. Therefore, we have provided the full name of the IMS- Ion Mobility Spectrometry and changed it in the text.
Comments 14: Mention to the table or figure represented the result of section 3.1.
Response 14: Thank you for pointing this out. We agree with this comment. Therefore, we have already referred to tables to corroborate the conclusions of section 3.1.
Line 274-278: Concurrently, as shown in Table 4 lactic acid bacteria generates numerous flavour-active compounds during fermentation, including acetaldehyde, acetone, acetyl, diacetyl, and acetic acid, imparting the distinctive flavour profile of PPY.
Comments 15: Section 3.1.1; Discuss more on how protein content could increase during fermentation.
Response 15: Thank you for pointing this out. Therefore, this may be due to some proteases produced by lactic acid bacteria during their growth and proliferation. It is also possible that the protein content increases due to the precipitation of whey at the end of fermentation, resulting in a decrease in water while the total amount of protein re-mains unchanged.
Line 279-287: Protein content is a critical indicator of PPY's nutritional value, significantly influencing both its quality and health benefits. As shown in Fig. 1, the protein content increased with the increase in fermentation time. This may be due to the precipitation of whey at the end of fermentation, which leads to a decrease in water while the total amount of proteins remains the same, resulting in an increase in protein content. This suggests that fermentation promotes higher protein levels, thereby enhancing the nutritional profile of pea protein PPY.
Comments 16: Figure 1 did not show protein concentration at 0 h.
Response 16: Thank you for pointing this out. We agree with this comment. The protein content at 0h was essentially the same as that at 2h. Therefore, we have recorrected Figure 1 and added the data at 0h.
Line 288-289:
Comments 17: Section 3.1.2; Discuss more on how fermentation could reduce trypsin inhibitor.
Response 17: Thank you for pointing this out. There are two reasons for this, one is that the acidic environment created by a drop in pH can denature and inactivate trypsin inhibitors. The second reason is that some of the proteases produced by lactic acid bacteria during growth and proliferation can hydrolyze the peptide bonds of trypsin inhibitors, thus destroying their spatial structure and inactivating them.
Line 299-303: There are two reasons for this, one is that the acidic environment created by a drop in pH can denature and inactivate trypsin inhibitors [29]. The second reason is that some of the proteases produced by lactic acid bacteria during growth and proliferation can hydrolyze the peptide bonds of trypsin inhibitors, thus destroying their spatial structure and inactivating them [30]. This indicates that fermentation effectively reduces trypsin inhibitor activity, enhancing the nutritional safety of pea protein PPY.
References:
[29] Vagadia, B.H.; Vanga, S.K.; Raghavan, V. Inactivation methods of soybean trypsin inhibitor - A review. Trends in Food Science & Technology 2017, 64, 115-125, doi:10.1016/j.tifs.2017.02.003.
[30]Luo, Z.; Zhu, Y.; Xiang, H.; Wang, Z.; Jiang, Z.; Zhao, X.; Sun, X.; Guo, Z. Advancements in Inactivation of Soybean Trypsin Inhibitors. Foods 2025, 14, doi:10.3390/foods14060975.
Comments 18: Section 3.1.4; Clarify the role of pH 6 during yogurt fermentation.
Response 18: Thank you for pointing this out. Although the isoelectric point of pea protein has not been reached at pH 6, the acidic environment has begun to expose the hydrophobic regions hidden inside the molecule to exposure and hydrophobic interactions, causing the protein to begin pre-aggregation and micelle-like aggregation behavior.
Line 375-381: After 4 h, the PPY gradually formed a gel, because the pH of the PPY all dropped below 6 one after another. Although the isoelectric point of pea protein has not been reached at pH 6, the acidic environment has begun to expose the hydrophobic regions hidden inside the molecule to exposure and hydrophobic interactions, causing the protein to begin pre-aggregation and micelle-like aggregation behavior.
Comments 19: There is no section 3.1.5.
Response 19: Thank you for pointing this out. Section 3.1.5 is located on line 364-390.
Comments 20: Section 3.1.7; Mention to amino acid contained in pea protein which could form disulfide-bond.
Response 20: Thank you for pointing this out. We agree with this comment. Therefore, we have added the amino acid that forms disulfide bonds - cysteine.
Line 417-419: Under acidic conditions, free sulfhydryl groups within cysteine are released and subsequently oxidized to disulfide bonds. This oxidation process reduces free sulfhydryl group content (Fig. 4A).

Reviewer 2 Report
Comments and Suggestions for Authors
I am very grateful for the opportunity to review the manuscript foods-3892866 by Zhang and coauthors. In this study, five fermentation systems were systematically investigated to elucidate the fermentation mechanisms of pea yoghurt and to explore effective strategies for eliminating undesirable soy-like flavours. The work is interesting but requires adjustments to improve the overall quality of the manuscript.
Comments:
Abstract
- Lines 17–20: Present more clearly the advantages of using pea proteins for yoghurt production compared to conventional dairy-based systems.
- Provide a brief step-by-step description of the production process, including formulations, concentrations, and the evaluations performed.
- Lines 29–31: Specify the parameters that yielded the best results.
-Lines 34–36: State the conclusions more clearly, including what improvements are still required to enhance the properties of the developed product.
- Replace repeated keywords with synonyms or alternative expressions not already included in the title.
- Lines 45–46: The environmental aspects require a clearer discussion. Plant proteins also have environmental impacts; therefore, it is necessary to be more precise, for example, regarding the carbon footprint of these raw materials.
- Line 50: Provide more accurate information, considering that some plant proteins may have allergenic potential.
- Lines 53–55: In addition to advantages, include the limitations of using such ingredients, since no component is entirely positive.
- Lines 53–55: Indicate the main markets and producers of peas worldwide.
- Lines 94–95: Use italics appropriately.
- Line 99: Remove the symbol “[”.
- Line 118: Is “post-cooking” the most suitable term for this step? Please confirm.
- Line 129: Confirm whether “8000 rpm” is correct.
- Line 130: Denote this as Equation 1 and update the numbering throughout the manuscript accordingly.
- Line 209: Clarify whether “10000” refers to rpm or ×g.
- Line 217 and throughout: It is not clear whether the treatments were applied to the ingredient, to the yoghurt base prior to fermentation, or to the final fermented product. Please describe more clearly.
- Lines 218–221: Check sentence structure and improve clarity.
- Line 257: Deepen the metabolic discussion regarding both Lactobacillus bulgaricus and Streptococcus thermophilus. How do they act in this type of product?
- Lines 258–265: Add appropriate references.
- Line 270: Clarify the meaning of the statement “promotes higher protein levels.”
- Line 273: In the figure legend, specify that red bars represent protein and grey bars represent trypsin inhibitors.
- Lines 281–283: The mechanism of reduction of inhibitors is unclear. Please describe in detail what causes this reduction.
- Lines 313–317: The explanation regarding changes in the volatile profile should be expanded to better clarify metabolic alterations, component degradation, etc. This should also be linked to what is described in lines 322–324.
- Lines 120, 345, 361, and throughout: The more appropriate term is “syneresis” (the opposite of how it is currently presented).
- Figure 6A: Resolution is too low. Please adjust to meet journal standards.
- Figure 6: Replace codes with treatment names to facilitate comparison.
- Lines 480–483: When applying the treatment to yoghurt, the authors should further discuss the issue of microbial inactivation. In many countries, fermented products must contain a minimum viable concentration of microorganisms to exert beneficial health effects. In this sense, it should also be discussed whether the treatment would be more effective if applied to the ingredient or to the pre-fermentation base.
- Line 557: Standardize the references according to the journal’s guidelines.
- Lines 648–650: It does not appear that optimal conditions were selected. Fermentation seems to have been carried out exactly as described in section 2.3, without optimization or condition selection.
Author Response
Comments 1: Lines 17–20: Present more clearly the advantages of using pea proteins for yoghurt production compared to conventional dairy-based systems.
Response 1: Thank you for pointing this out. We agree with this comment. Therefore, we have already explained in the article the advantages of pea protein yogurt ingredients that are green, lactose cholesterol free and adapted to lactose intolerant people.
Line 17-19: Pea protein yogurt (PPY), as an alternative to traditional dairy yogurt, has the advantages of green raw materials, lactose cholesterol-free, and adapted to the needs of lactose intolerant people.
Comments 2: Provide a brief step-by-step description of the production process, including formulations, concentrations, and the evaluations performed.
Response 2: Thank you for pointing this out. We agree with this comment. Therefore, we have already briefly described the production process in the article.
Line 19-22: PPY was prepared by fermenting a mixture of pea protein and water (1:10, w/v) supplemented with 5% fructose for 10 h after heat sterilization. During fermentation, lactic acid bacteria metabolize pea protein to produce aldehydes and other aromatic compounds, imparting a unique sweet-sour balance and mellow flavour.
Comments 3: Lines 29–31: Specify the parameters that yielded the best results.
Response 3: Thank you for pointing this out. We agree with this comment. Therefore, we have added the optimal parameters in the text. However, I would like to explain here that our autoclave treatment is carried out in an autoclave, which is not able to regulate the pressure, and therefore our optimal parameters relate only to temperature and time.
Line 1-2: Furthermore, high-temperature and high-pressure treatments (121℃, 3min) demon-strated superior effectiveness in reducing soybean-like flavours.
Comments 4: Lines 34–36: State the conclusions more clearly, including what improvements are still required to enhance the properties of the developed product.
Response 4: Thank you for pointing this out. We agree with this comment. Therefore, we have revised the manuscript to state the conclusions more clearly. The main conclusions of our study are that this research provides both theoretical support and practical processing parameters for flavor modulation in pea protein yogurt, and that further formulation optimization is still required to improve its nutritional and textural properties.
Line 37-42: These findings provide a theoretical basis and processing parameters for flavor modulation in PPY; however, further formulation optimization is required to enhance its nutritional and textural properties. PPY shows promise as a potential alternative to conventional dairy products in the future.
Comments 5: Replace repeated keywords with synonyms or alternative expressions not already included in the title.
Response 5: Thank you for pointing this out. We agree with this comment. Therefore, we have changed bean smell removal to bean odor removal method.
Comments 6: Lines 45–46: The environmental aspects require a clearer discussion. Plant proteins also have environmental impacts; therefore, it is necessary to be more precise, for example, regarding the carbon footprint of these raw materials.
Response 6: Thank you for pointing this out. We agree with this comment. Therefore, we We have characterized the environmental aspect more precisely. Livestock now leads all sectors in carbon emissions and has a greater environmental impact.
Line 51-55: On one hand, carbon emissions have risen with the rapid expansion of the livestock sector, which has now jumped to the top of all sectors; on the other hand, approximately two-thirds of the global population experiences lactose intolerance.
Comments 7: Line 50: Provide more accurate information, considering that some plant proteins may have allergenic potential.
Response 7: Thank you for pointing this out. We agree with this comment. Therefore, we didn't check out the production of plant proteins with allergy potential, so we removed the allergy section here and changed it to provide a new option for lactose intolerant people.
Line 57-59: Plant-based proteins, produced globally in large quantities (approximately 4.2 million tonnes per year), are lactose-free, and their products offer a new option for lactose-intolerant people [2,3].
Comments 8: Lines 53–55: In addition to advantages, include the limitations of using such ingredients, since no component is entirely positive.
Response 8: Thank you for pointing this out. We agree with this comment. Therefore, we have added in the text the limitations of peas in the food sector, including a pronounced pea odor and poor solubility and gelation properties.
Line 61-71: Notably, it is noteworthy that peas are cultivated on a large scale globally, with Russia alone producing approximately 34 million tons annually—accounting for one-third of the world’s total pea production. The Asia-Pacific region has emerged as the fastest-growing market for pea protein consumption, a trend propelled by its large population base and rising overall demand for protein. These significant outputs have garnered considerable attention from both industry stakeholders and researchers. However, peas have defects such as pronounced pea odor, poor solubility and gelation in food applications, which together limit the development of peas in the food industry.
Comments 9: Lines 53–55: Indicate the main markets and producers of peas worldwide.
Response 9: Thank you for pointing this out. We agree with this comment. Therefore, we have indicated the main producer of peas in the world (Russia) and the market with the fastest growing demand (Asia-Pacific market) and reorganized the paragraph.
Line 61-71: Notably, it is noteworthy that peas are cultivated on a large scale globally, with Russia alone producing approximately 34 million tons annually—accounting for one-third of the world’s total pea production. The Asia-Pacific region has emerged as the fastest-growing market for pea protein consumption, a trend propelled by its large population base and rising overall demand for protein. These significant outputs have garnered considerable attention from both industry stakeholders and researchers. However, peas have defects such as pronounced pea odor, poor solubility and gelation in food applications, which together limit the development of peas in the food industry.
Comments 10: Lines 94–95: Use italics appropriately.
Response 10: Thank you for pointing this out. We agree with this comment. Therefore, we have made changes.
Line 109-110: Pea protein powder, purchased from Yantai Shuangta Foodstuffs Co. Ltd; leavening agent was a mixed compound of Lactobacillus bulgaricus and Streptococcus thermophilus purchased from Yiran Biological Co. Ltd. of Shijiazhuang City, Hebei Province, China;
Comments 11: Line 99: Remove the symbol “[”.
Response 11: Thank you for pointing this out. We agree with this comment. Therefore, we have removed the symbol “[”.
Comments 12: Line 118: Is “post-cooking” the most suitable term for this step? Please confirm.
Response 12: Thank you for pointing this out. We agree with this comment. Therefore, we have replaced “post-cooking” with “post-fermentation ripening”.
Comments 13: Line 129: Confirm whether “8000 rpm” is correct.
Response 13: Thank you for pointing this out. We agree with this comment. Therefore, we have changed “8000 rpm” to “8000 ×g”.
Comments 14: Line 130: Denote this as Equation 1 and update the numbering throughout the manuscript accordingly.
Response 14: Thank you for pointing this out. We agree with this comment. Therefore, we have revised it.
Comments 15: Line 209: Clarify whether “10000” refers to rpm or ×g.
Response 15: Thank you for pointing this out. We agree with this comment. Therefore, we have modified it to "10000×g".
Comments 16: Line 217 and throughout: It is not clear whether the treatments were applied to the ingredient, to the yoghurt base prior to fermentation, or to the final fermented product. Please describe more clearly.
Response 16: Thank you for pointing this out. We agree with this comment. Treatment applied before fermentation. We have made changes in the appropriate places and re-described them in the subsequent sections.
Line 236-237: Different treatments before fermentation and GC-IMS Processing Conditions.
Comments 17: Lines 218–221: Check sentence structure and improve clarity.
Response 17: Thank you for pointing this out. We agree with this comment. Therefore, we have redescribed this paragraph as follows.
Line 238-244: 2.12.1. Different treatments before fermentation
As shown in Table 2, the preparation process of A2 was consistent with that of Section 2.3; A3 changed the heating of the wall-breaker in Section 2.3 to microwave heating (500w, 5min); A4 changed the heating of the wall-breaker in Section 2.3 to autoclaving (121°C, 3min); A5 was based on the enzyme digestion of A2 prior to heating in the wall-breaker (Alcohol dehydrogenase: 0.5%, aldehyde dehydrogenase: 0.5%, 2h); A6 was to replace the ingredients with a 1:1 blend of pea protein powder and coconut protein powder, and the other treatments were the same as A2.
Comments 18: Line 257: Deepen the metabolic discussion regarding both Lactobacillus bulgaricus and Streptococcus thermophilus. How do they act in this type of product?
Response 18: Thank you for pointing this out. We agree with this comment. Therefore, we have added two fermentation advantages of lactobacilli to the article, see below for details.
Line 287-297: The combination of Streptococcus thermophilus with Lactobacillus bulgaricus subsp. bulgaricus has been shown to produce yogurt with lower synergistic effects, better texture, and superior organoleptic qualities compared to other fermenters in yogurt [24]. After inoculation of Lactobacillus bulgaricus with Streptococcus thermophilus in pea protein solution, they started to utilize the available nutrients for growth and proliferation, producing new proteases. During metabolism, organic acids are rapidly produced, which gradually lowers the pH of the solution, and the proteins begin to pre-aggregate, promoting the formation of gel structures.
Comments 19: Lines 258–265: Add appropriate references.
Response 19: Thank you for pointing this out. We agree with this comment. Therefore, we have added references.
References:
[24] Han, X.; Yang, Z.; Jing, X.; Yu, P.; Zhang, Y.; Yi, H.; Zhang, L. Improvement of the Texture of Yogurt by Use of Exopolysaccharide Producing Lactic Acid Bacteria. BioMed Research International 2016, 2016, 1-6, doi:10.1155/2016/7945675.
[25] Damodar, D.; Gaurav, K.; Lavaraj, D.; Dinesh, S.; Sushil, D. The choice of probiotics affects the rheological, structural, and sensory attributes of lupin-oat-based yoghurt. Food Hydrocolloids 2024, 156, 110353-110353, doi:10.1016/j.foodhyd.2024.110353.
Comments 20: Line 270: Clarify the meaning of the statement “promotes higher protein levels.”
Response 20: Thank you for pointing this out. We agree with this comment. The total number of grams in the pea protein is unchanged, and the increase in content is due to the fact that the whey continues to precipitate out during the fermentation process, resulting in a decrease in water and an increase in protein content.
Line 305-312: As shown in Fig. 1, the protein content increased with the increase in fermentation time. This may be due to the precipitation of whey at the end of fermentation, which leads to a decrease in water while the total amount of proteins remains the same, resulting in an increase in protein content. This suggests that fermentation promotes higher protein levels, thereby enhancing the nutritional profile of PPY.
Comments 21: Line 273: In the figure legend, specify that red bars represent protein and grey bars represent trypsin inhibitors.
Response 21: Thank you for pointing this out. We agree with this comment. Therefore, we redrew a diagram with a legend.
Line 314-315:
Comments 22: Lines 281–283: The mechanism of reduction of inhibitors is unclear. Please describe in detail what causes this reduction.
Response 22: Thank you for pointing this out. We agree with this comment. Therefore, we We describe the reasons for the decrease in inhibitors. There are two reasons for the decrease in inhibitors. One is that the acidic environment created by a drop in pH can denature and inactivate trypsin inhibitors. The second reason is that some of the proteases produced by lactic acid bacteria during growth and proliferation can hydrolyze the peptide bonds of trypsin inhibitors, thus destroying their spatial structure and inactivating them.
Line 323-329: There are two reasons for this, one is that the acidic environment created by a drop in pH can denature and inactivate trypsin inhibitors. The second reason is that some of the proteases produced by lactic acid bacteria during growth and proliferation can hydrolyze the peptide bonds of trypsin inhibitors, thus destroying their spatial structure and inactivating them. This indicates that fermentation effectively reduces trypsin inhibitor activity, enhancing the nutritional safety of PPY.
Comments 23: Lines 313–317: The explanation regarding changes in the volatile profile should be expanded to better clarify metabolic alterations, component degradation, etc. This should also be linked to what is described in lines 322–324.
Response 23: Thank you for pointing this out. We agree with this comment. Therefore, we have added some explanations as shown below.
Line 359-365: This reduction is likely due to the inactivation of lipoxygenase during heating, which pre-vents the oxidation of fatty acids that generate these volatile compounds, and due to the active enzyme system of lactic acid bacteria during fermentation, e.g. aldehyde dehydrogenase converts the undesirable flavor hexanal to hexanoic acid. Meanwhile, fer-mentation led to an increase in compounds characteristic of fermented foods, such as 3-methylbutyric acid and isobutyric acid, as well as pleasant aroma components including 1-hexanol, 1-heptanol, and 2-pentanone.
Comments 24: Lines 120, 345, 361, and throughout: The more appropriate term is “syneresis” (the opposite of how it is currently presented).
Response 24: Thank you for pointing this out. We agree with this comment. Therefore, we have changed it to “syneresis”.
Comments 25: Figure 6A: Resolution is too low. Please adjust to meet journal standards.
Response 1: Thank you for pointing this out. We agree with this comment. Therefore, we readjusted the picture.
Comments 26: Figure 6: Replace codes with treatment names to facilitate comparison.
Response 26: Thank you for pointing this out. We agree with this comment. Therefore, we have labeled the treatments in the figure.
Comments 27: Lines 480–483: When applying the treatment to yoghurt, the authors should further discuss the issue of microbial inactivation. In many countries, fermented products must contain a minimum viable concentration of microorganisms to exert beneficial health effects. In this sense, it should also be discussed whether the treatment would be more effective if applied to the ingredient or to the pre-fermentation base.
Response 27: Thank you for pointing this out. We agree with this comment. All our treatments are carried out before fermentation, and the specific procedures have been re-explained in detail in the previous section, so our yogurt is a live-bacteria type of yogurt, and does not involve microbial inactivation after fermentation.
Comments 28: Line 557: Standardize the references according to the journal’s guidelines.
Response 28: Thank you for pointing this out. We agree with this comment. Therefore, we have reformatted the references in accordance with journal requirements.
Comments 29: Lines 648–650: It does not appear that optimal conditions were selected. Fermentation seems to have been carried out exactly as described in section 2.3, without optimization or condition selection.
Response 29: Thank you for pointing this out. We agree with this comment. The process parameters used in this post are based on the best process parameters that I have previously, with reference to physicochemical, textural and nutritional to optimize. This post focuses on flavor and therefore does not describe the pre-optimization of the process. Thank you for your understanding.

Reviewer 3 Report
Comments and Suggestions for Authors
The manuscript explores the fermentation mechanisms of pea protein yogurt, focusing on gel formation through hydrophobic interactions and disulfide bonds, changes in protein content during fermentation, and the profiling of 43 volatile compounds using GC-IMS. It also examines methods to reduce undesirable bean-like flavors, such as hexanal and 2-pentylfuran, by using physical treatments (microwave and high-temperature/high-pressure), bio-enzymatic methods (double-enzyme hydrolysis), and masking techniques (coconut protein complexing). The main contributions include understanding how fermentation affects flavor and texture, as well as evidence for combined deodorization effects, which have practical value in improving the sensory and textural qualities of plant-based yogurt alternatives. The study’s strengths lie in its comprehensive analysis of volatile compounds and intermolecular interactions, addressing key challenges for the commercial development of pea-based products.
General Comments
This study examines how different processing methods affect fermentation in pea protein yogurt, a topic of considerable interest and importance in the context of sustainable, lactose-free dairy alternatives. However, the manuscript has significant readability issues, particularly in clearly defining the samples and analysis steps, making the results and discussion difficult to follow. Major revisions are suggested, including a diagram of the experimental process and clearer descriptions of the samples used at each stage. Additionally, the discussion section would benefit from further references to recent studies, as it currently cites only 15 sources, limiting its depth. These changes would enhance clarity and scientific rigor.
Specific Comments
- Title: The title implies a focus on pea protein yogurt, but the text frequently references "soymilk," creating ambiguity. Specify the product clearly, if it is a pea protein product e.g., "Fermentation Mechanism of Pea Protein-Based Yogurt and Its Bean Odor Removal Methods," to avoid confusion with soy-derived products.
- Materials and Methods: This section is ineffectively structured and requires comprehensive rephrasing for clarity and conciseness. A schematic diagram illustrating the overall study design, including sample preparation, treatment applications, and analytical timelines, is essential to clarify the progression from pea protein isolate to final yogurt variants. Explicitly describe which samples are analyzed at each step, as the current narrative obscures this, contributing to confusion in subsequent sections.
- Line 94: Italicize microorganism names (e.g., Lactobacillus bulgaricus and Streptococcus thermophilus). Additionally, specify the exact strain(s) and/or starter used, as this is not detailed.
- Lines 98-102: Repetitive phrasing and apparent typographical errors (e.g., "Aladdin; and all of the usedAll other chemicals used were at least analytical 98 grade (Beijing Chemical Reagent Co., Ltd., Beijing, China).[All other chemicals used were 99 at least of analytical grade (Beijing Chemical Reagent Co.,") should be corrected. Rephrase Section 2.1 entirely for conciseness.
- Section 2.2: Please rephrase to make it clearer.
- Line 113: Provide detailed homogenization parameters (e.g., pressure in MPa, cycles, temperature) to ensure reproducibility. Note that "soymilk" is mentioned here without prior definition in materials; replace with "pea protein beverage" or define explicitly to maintain consistency with pea protein focus. The manuscript inconsistently alternates between soy and pea yogurt terminology, making it difficult to discern the primary subject—resolve this via consistent nomenclature and the suggested diagram.
- Line 130: Number the WHC equation (e.g., Equation (1)) in accordance with the journal's author instructions.
- Line 139: Rephrase for clarity: "Protein content was quantified using the Kjeldahl method as described by Wang et al. [19], with total nitrogen multiplied by a conversion factor of 6.25."
- Table 1: Reference the table explicitly in the text (e.g., "as detailed in Table 1"). Remove any yellow highlighting or labels, as they appear extraneous (similarly for annotations at Line 218).
- Line 192: Format the free sulfhydryl equation per journal guidelines, e.g., as Equation (2).
- Lines 193-194: Correct units for the molar extinction coefficient (e.g., 73.53 L mol⁻¹ cm⁻¹) and molar absorptivity (1.36 × 10⁴ M⁻¹ cm⁻¹) to align with standard spectroscopic conventions.
- Results and Discussion: Expand the discussion with comparisons to analogous studies on plant-based yogurt fermentation and off-flavor mitigation, incorporating additional recent references to contextualize findings (e.g., microbial dynamics in pea vs. soy systems). Currently, the limited citations (15 total) weaken interpretive depth.
- Section 3.1: Support factual statements with references to your results or external studies.
- Lines 258-265: Several claims on fermentation mechanisms (e.g., pH-driven aggregation and volatile production) lack supporting references, sources, or direct evidence from your data—provide citations or link to specific results (e.g., pH profiles in Figure 3).
- Line 267: Please add a reference for this fact.
- Figure 1: Include "protein content" explicitly in the caption for clarity.
- Figure 3: Specify "titratable acidity" in the caption.
- Line 354: Clarify the statement on pH decline after 7 h: Specify the groups compared (e.g., control vs. treated) and reconcile with the figure showing only one sample trace. If multi-group data exist, update the figure accordingly; otherwise, revise to reflect single-sample analysis.
Author Response
Comments 1: Title: The title implies a focus on pea protein yogurt, but the text frequently references "soymilk," creating ambiguity. Specify the product clearly, if it is a pea protein product e.g., "Fermentation Mechanism of Pea Protein-Based Yogurt and Its Bean Odor Removal Methods," to avoid confusion with soy-derived products.
Response 1: Thank you for pointing this out. We agree with this comment. Our product is Pea Protein Yogurt. We have checked and corrected the text about peas and soybeans throughout the article, and have changed all of it to Pea Protein Yogurt (PPY) and corrected the soy milk to Pea Protein Solution (PPS).
Comments 2: Materials and Methods: This section is ineffectively structured and requires comprehensive rephrasing for clarity and conciseness. A schematic diagram illustrating the overall study design, including sample preparation, treatment applications, and analytical timelines, is essential to clarify the progression from pea protein isolate to final yogurt variants. Explicitly describe which samples are analyzed at each step, as the current narrative obscures this, contributing to confusion in subsequent sections.
Response 2: Thank you for pointing this out. We agree with this comment. Therefore, we have rewritten the Materials and Methods section to ensure that the article is readable and organized.
Comments 3: Line 94: Italicize microorganism names (e.g., Lactobacillus bulgaricus and Streptococcus thermophilus). Additionally, specify the exact strain(s) and/or starter used, as this is not detailed.
Response 3: Thank you for pointing this out. We agree with this comment. Therefore, we have italicized the microbial names and labeled the exact strains.
Comments 4: Lines 98-102: Repetitive phrasing and apparent typographical errors (e.g., "Aladdin; and all of the used All other chemicals used were at least analytical 98 grade (Beijing Chemical Reagent Co., Ltd., Beijing, China). [All other chemicals used were 99 at least of analytical grade (Beijing Chemical Reagent Co.,") should be corrected. Rephrase Section 2.1 entirely for conciseness.
Response 4: Thank you for pointing this out. We agree with this comment. Therefore, we have rewritten section 2.1.
Lines109-127: Pea protein powder was obtained from Yantai Shuangta Foodstuffs Co., Ltd. A mixed leavening agent consisting of Lactobacillus bulgaricus and Streptococcus thermophilus was supplied by Yiran Biological Co., Ltd. (Shijiazhuang City, Hebei Province, China). Sugar was purchased from a local supermarket in Beijing, China. The following ortho-ketones (all analytical grade) were sourced from Aladdin: 2-butanone, 2-pentanone, 2-hexanone, 2-heptanone, 2-octanone, and 2-nonanone. All other chemicals used were of analytical grade and obtained from Beijing Chemical Reagent Co., Ltd. (Beijing, China). High-purity nitrogen (99.999%) and 20 mL headspace vials were provided by Shandong Haineng Scientific Instrument Co. Separation was performed using an MXT-5 capillary column (15 m × 0.53 mm, 1.0 μm; Restek, USA).
Comments 5: Section 2.2: Please rephrase to make it clearer.
Response 5: Thank you for pointing this out. We agree with this comment. Therefore, we have rewritten section 2.2.
Line129-142: The following instruments were used: a high-pressure homogenizer (JXNANO-15, Shanghai); wall-breaker (BM9, Shenzhen, China); a vertical autoclave sterilizer (LDZF-50L-I, Shanghai); a thermostatic incubation oscillator (Z-intelligent HNY-301, Tianjin); a scanning electron microscope (Phenom XL, MVE 033310-0151-L); a microwave oven (G80F25MSLVII-ZN(MO), Guangdong); a wall-breaker (BM9, Shenzhen); a FlavourSpec® gas-phase ion mobility spectrometer (G.A.S., Dortmund, Germany); a CTC-PAL 3 static headspace autosampler (CTC Analytics AG, Zwingen, Switzerland); and VOCal data processing software (version 0.4.10, G.A.S., Dortmund, Germany).
Comments 6: Line 113: Provide detailed homogenization parameters (e.g., pressure in MPa, cycles, temperature) to ensure reproducibility. Note that "soymilk" is mentioned here without prior definition in materials; replace with "pea protein beverage" or define explicitly to maintain consistency with pea protein focus. The manuscript inconsistently alternates between soy and pea yogurt terminology, making it difficult to discern the primary subject—resolve this via consistent nomenclature and the suggested diagram.
Response 6: Thank you for pointing this out. We agree with this comment. Therefore, we have added specific parameters for high pressure homogenization to the text. And corrected the terminology by correcting soymilk to pea protein solution.
Lines 114-147: Pea protein powder was mixed with sterile water at a material-liquid ratio of 1:10, 5% fructose was added, high-pressure homogenization (60℃, 30~40MPa, single instantaneous processing) was carried out, and Pea protein solution (PPS) soymilk was made using a wall-breaker, a process that included a heating step.
Comments 7: Line 130: Number the WHC equation (e.g., Equation (1)) in accordance with the journal's author instructions.
Response 7: Thank you for pointing this out. We agree with this comment. Therefore, we have numbered all the formulas in the text.
Comments 8: Line 139: Rephrase for clarity: "Protein content was quantified using the Kjeldahl method as described by Wang et al. [19], with total nitrogen multiplied by a conversion factor of 6.25."
Response 8: Thank you for pointing this out. We agree with this comment. Therefore, we have made changes here.
Comments 9: Table 1: Reference the table explicitly in the text (e.g., "as detailed in Table 1"). Remove any yellow highlighting or labels, as they appear extraneous (similarly for annotations at Line 218).
Response 9: Thank you for pointing this out. We agree with this comment. Therefore, we have cited this table in the text and removed all yellow highlighting and labeling.
Comments 10: Line 192: Format the free sulfhydryl equation per journal guidelines, e.g., as Equation (2).
Response 10: Thank you for pointing this out. We agree with this comment. Therefore, we have modified this to Equation (2).
Comments 11: Lines 193-194: Correct units for the molar extinction coefficient (e.g., 73.53 L mol⁻¹ cm⁻¹) and molar absorptivity (1.36 × 10⁴ M⁻¹ cm⁻¹) to align with standard spectroscopic conventions.
Response 11: Thank you for pointing this out. We agree with this comment. Therefore, we have made changes to the unit.
Comments 12: Results and Discussion: Expand the discussion with comparisons to analogous studies on plant-based yogurt fermentation and off-flavor mitigation, incorporating additional recent references to contextualize findings (e.g., microbial dynamics in pea vs. soy systems). Currently, the limited citations (15 total) weaken interpretive depth.
Response 12: Thank you for pointing this out. We agree with this comment. Therefore, we have expanded the discussion and introduced additional references.
Comments 13: Section 3.1: Support factual statements with references to your results or external studies.
Response 13: Thank you for pointing this out. We agree with this comment. Therefore, we have added references here.
References:
[24] Han, X.; Yang, Z.; Jing, X.; Yu, P.; Zhang, Y.; Yi, H.; Zhang, L. Improvement of the Texture of Yogurt by Use of Exopolysaccharide Producing Lactic Acid Bacteria. BioMed Research International 2016, 2016, 1-6, doi:10.1155/2016/7945675.
[25] Damodar, D.; Gaurav, K.; Lavaraj, D.; Dinesh, S.; Sushil, D. The choice of probiotics affects the rheological, structural, and sensory attributes of lupin-oat-based yoghurt. Food Hydrocolloids 2024, 156, 110353-110353, doi:10.1016/j.foodhyd.2024.110353.
Comments 14: Lines 258-265: Several claims on fermentation mechanisms (e.g., pH-driven aggregation and volatile production) lack supporting references, sources, or direct evidence from your data—provide citations or link to specific results (e.g., pH profiles in Figure 3)
Response 14: Thank you for pointing this out. We agree with this comment. Therefore, we have added references here. and linked to specific results (pH curves in Figure 3).
Comments 15: Line 267: Please add a reference for this fact.
Response 15: Thank you for pointing this out. We agree with this comment. Therefore, we have added references here.
Comments 16: Figure 1: Include "protein content" explicitly in the caption for clarity.
Response 16: Thank you for pointing this out. We agree with this comment. Therefore, we have made changes here.
Comments 17: Figure 3: Specify "titratable acidity" in the caption.
Response 17: Thank you for pointing this out. We agree with this comment. Therefore, we have made changes here.
Comments 18: Line 354: Clarify the statement on pH decline after 7 h: Specify the groups compared (e.g., control vs. treated) and reconcile with the figure showing only one sample trace. If multi-group data exist, update the figure accordingly; otherwise, revise to reflect single-sample analysis.
Response 18: Thank you for pointing this out. First, I'll explain what is in Figure 3. The black line in Figure 3A is the pH curve and the red line is the titratable acidity curve, both curves are for the same sample. Secondly, the two curves are put together to make it more obvious that pH and titratable acidity are inversely proportional as the fermentation time increases. We hope to have your understanding and support.
Round 2
Reviewer 2 Report
Comments and Suggestions for Authors
The article has been modified following the suggested corrections and has been substantially revised to align with the required standards.
Reviewer 3 Report
Comments and Suggestions for Authors
Dear Authors, thank you for thoroughly addressing my comments and incorporating the suggested revisions into the manuscript. I am pleased to confirm that I am satisfied with the corrections made. Your thorough revisions have significantly strengthened the manuscript’s scientific rigor and clarity.
I appreciate your efforts and believe the manuscript is now well-suited for publication.